# Feather moult and bird appearance are correlated with global warming over the last 200 years

Y. Kiat [1], Y. Vortman[2] & N. Sapir [1]

Global warming alters various avian phenological processes, including advanced reproduction and migration schedules. In birds, individual appearance is largely determined by plumage, influencing, for example, bird attractiveness, social status and camouflage. Juveniles of most passerine species replace their nest-grown plumage during the first months of life, a process that is called post-juvenile feather moult. Using data from ten natural history collections, we show that the extent of the post-juvenile moult has increased significantly over the last 212 years (1805–2016), a trend that is positively correlated with the temperature of the environment. Therefore, it seems that birds replaced more feathers under warmer conditions, causing juveniles to appear more similar to adult birds. Moreover, in several species, we describe a male–female switch in the extent of moult, with females currently replacing more feathers than males compared to the past. These results demonstrate different biological responses to climate warming by different phenotypes.

---

[1] Animal Flight Laboratory, Department of Evolutionary and Environmental Biology and the Institute of Evolution, University of Haifa, 3498838 Haifa, Israel. [2] Hula Research Centre, Department of Animal Sciences, Tel-Hai College, 1220800 Upper Galilee, Israel. Correspondence and requests for materials should be addressed to Y.K. (email: yosefkiat@gmail.com) or to Y.V. (email: vortmanyo@gmail.com) or to N.S. (email: nirs@sci.haifa.ac.il)

Global warming alters various phenological processes and consequently may disrupt phenological matches in many populations, species and ecosystems[1–5]. However, due to incomplete knowledge of the mechanisms linking temperature change and ecological responses, life-history alterations and evolutionary adaptations, predicting future responses of biological systems to global warming is still a major challenge[3,5–7].

Feather moult is one of the most important and energy-demanding life history processes in the avian annual cycle[8]. Feather renewal is essential for maintaining plumage utility, because feathers accumulate wear and tear, and consequently their functionality in thermoregulation, aerodynamics and ornamentation deteriorates with time. In most passerine species, juveniles that replace their nest-grown feathers gain the appearance of adult plumage following their first feather moult. However, not all juvenile feathers are replaced at this stage, with the extent of partial moult depending mainly on available time and food resources[9,10]; accordingly, there are implications for bird behaviour, morphology and ecological interactions[11–13]. For example, the extent of the post-juvenile moult may determine the appearance of the individual, influencing its attractiveness, social status and camouflage.

Environmental conditions, including ambient temperatures, are known to influence available time and resources for moult, and may affect the extent of moult and its duration[9,10]. In spite of demonstrated effects of global warming on avian biology[1,3,14,15] and numerous studies revealing differences in thermoregulation[16,17] and behaviour[18,19] between females and males, whether the warming climate influences feather moulting and whether the two sexes respond similarly are still open questions. To date, only two single-species and relatively short-term studies (33 and 11 years) demonstrated how feather moult advanced in response to climate warming[4,20], with no reported effects on moult extent or differences between female and male birds. In this study, we show that the extent of the post-juvenile moult has increased significantly over the past 212 years (1805–2016), a trend that is positively correlated with the temperature of the environment. This suggests that birds replaced more feathers under warmer conditions, and causing juveniles to appear more similar to adult birds. Moreover, in several species, we find a male–female switch in moult extent, with females currently replacing more feathers than males compared to the past.

## Results

**Feather moult extent and climate warming.** We measured the extent of feather moult in 19 passerine bird species that breed in the Western Palearctic ecozone and that have various migration strategies ($n = 4012$ individuals; Supplementary Table 1 and Supplementary Fig. 1) using skin specimens stored in ten natural-history museum collections, and additionally field measured live individuals in several countries (see details in Supplementary Table 2). We found that over the past 212 years (1805–2016), partial moult of juveniles became more extensive, correlating with global temperature increase [global mean temperature anomalies (GMTA; see Methods); Figs. 1a and 2a]. In 16 out of the 19 species examined, our models show a positive effect of GMTA on moult extent [Akaike information criterion, modified for small sample sizes (AICc); ΔAICc ≥ 2.10; Supplementary Table 3, see also Supplementary Table 4 for the effect of the year]. By applying phylogenetic generalised least square (PGLS) regression using global bird phylogeny[21], we found that bird response to GMTA is indistinguishable between species exhibiting short- ($n = 12$ species) and long-distance ($n = 7$ species) migration, as well as between species with pre-migration moult (in the breeding areas; $n = 15$ species) and post-migration moult (after arrival to the tropical wintering areas; $n = 4$ species; Supplementary Table 5).

**Sex-dependent response to climate warming.** Among 10 sexually dichromatic species, we found a stronger response of females to GMTA than that of males in four species (Fig. 1b and Supplementary Table 3). In these species, the model that best explained the variation in moult extent included GMTA, individual sex and their interaction (ΔAICc ≥ 3.80). In these four species, the stronger response of females to warmer conditions than that of males caused a switch in the extent of feather moult between females and males, with females replacing more feathers than males from ~1990 onwards. In six other species, males and females did not differ in their response to GMTA (Supplementary Table 3). In the remaining nine species, males and females could not be distinguished and consequently the effect of bird sex could not be examined.

**The effect of sexual-dichromatism level.** We furthermore found that in the 10 sexually dichromatic species, the level of sexual-dichromatism in plumage ornamentation affects the magnitude of the difference in their sex-dependent response to GMTA. In species that are characterised by more ornamented male feathers compared with female feathers (*e.g.*, Desert Wheatear *Oenanthe deserti* and Collared Flycatcher), the response of females to GMTA is stronger ($r^2 = 0.41$, $F_{1,8} = 5.50$, $P = 0.047$, $λ = 0.00$; PGLS), while in species with relatively weak sexual dichromatism, the response of males and females to GMTA was similar such that their moult extent increased similarly for the two sexes with increasing GMTA (Fig. 3).

## Discussion

Among most Western Palearctic passerines, the main moulting period occurs immediately after the end of the breeding season, usually near the breeding grounds. Consequently, the birds have sufficient time to complete the moult process before food is depleted during the winter. Other birds, consisting several long-distance migrant species, moult later, such that the replacement of their feathers occurs immediately after their arrival to the tropical wintering areas. Four important phenological changes in the avian annual cycle were reported as widespread responses to global warming: (I) Earlier spring arrival to the breeding grounds among both short- and long-distance migratory passerines[4,22]; (II) earlier breeding timing (e.g., egg-laying and fledging) among residents as well as short- and long-distance migratory passerines[3,4]; (III) delayed onset of autumn migration, mainly among short-distance migratory passerines, which may be caused by reduced migration distance[15,22,23]; and (IV) earlier autumn migration mainly among long-distance migratory passerines[15]. These changes affect the scheduling of breeding, moulting and migration–all highly energy-demanding annual cycle processes[4,24,25].

Delayed autumn migration or earlier breeding may result in a longer pre-migration moult period by advancing the moult onset[4] and, likely, also delaying its termination. This extension of the moult period could be advantageous to birds that moult before migration (short- or long-distance migrants; Fig. 2b), allowing the birds to replace more feathers and hence to increase their moult extent. For long-distance migrants that moult in the tropical wintering areas, the advancement of the autumn migration[15] and the earlier migration onset may result in an earlier arrival to the tropical wintering areas, which might prolong the time available for moult immediately after the autumn migration (Fig. 2b). Moreover, earlier moult after the autumn migration in the northern tropics which are characterised by summer rains could be advantageous due to the higher food abundance in this region in early autumn, which nonetheless declines soon afterwards[26,27]. The earlier arrival and overall extension of the time available for

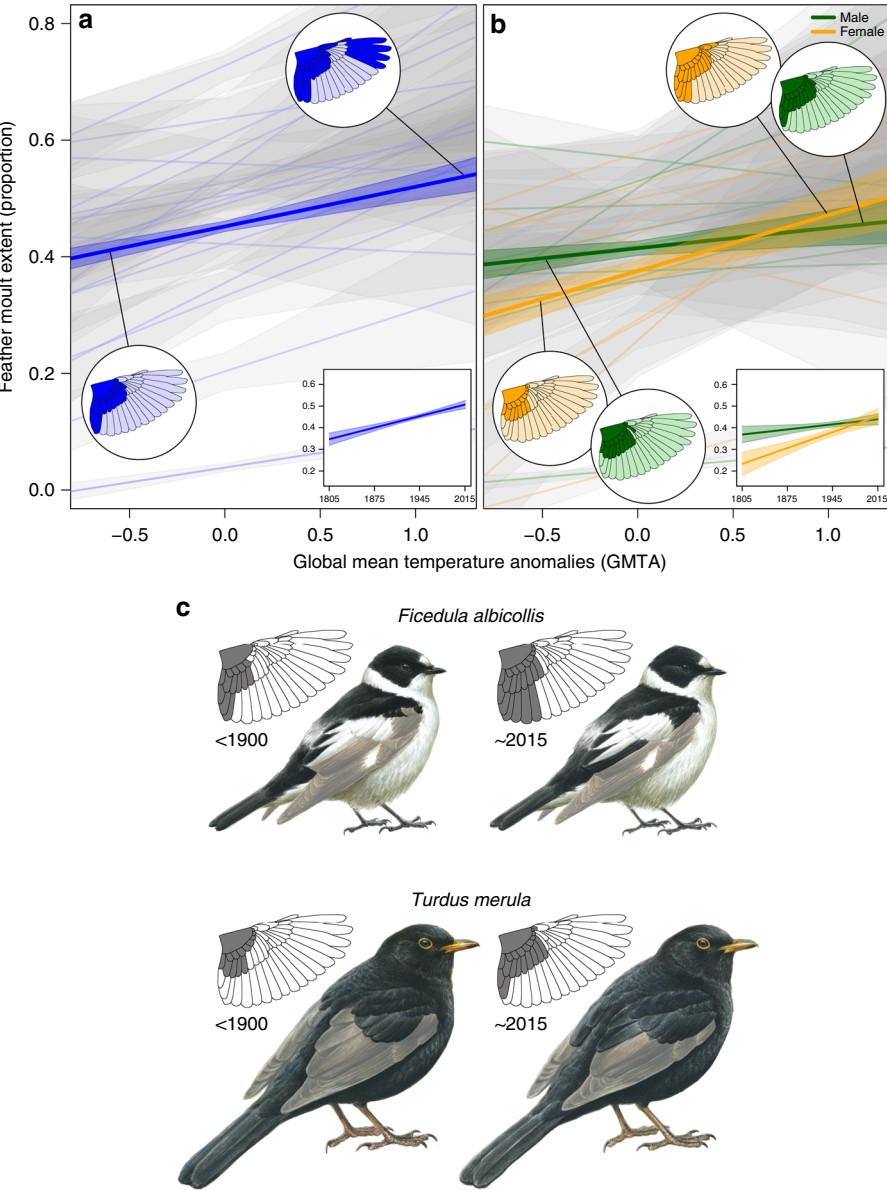

**Fig. 1** The relationship between global mean temperature anomalies (GMTA) and feather moult extent. **a** The extent of post-juvenile moult of 19 passerine species, shown separately for each species (thin lines ± 95% confidence intervals, CIs) and the general trend (bold line ± CIs). The trend indicates a significant increase of moult extent with increasing GMTA in 16 species (ΔAICc ≥ 2.10; Supplementary Table 3). The two circled insets depict the extent of wing moult in the beginning and the end of the examined period (1805–2016). The boxed inset illustrates the general trend of moult extent across the study period (years). **b** Male and female moult extent of 10 sexually dichromatic passerine species throughout the examined period, shown separately for each species (thin lines ± CIs) and the general trend (bold line ± CIs). In four species, the results indicate a stronger response to warming climates in females than in males (ΔAICc ≥ 3.80), leading to an overall more extensive moult of females than that of males since ~1990. Thus, in these species, females replaced more feathers than males under warmer conditions. **c** Two examples demonstrating how the increasing extent of moult changed the appearance of a first year male Collared Flycatcher *Ficedula albicollis* (top two illustrations) and Eurasian Blackbird *Turdus merula* (bottom two illustrations) before 1900 (left) compared to 2015 (right). Since the birds moulted more feathers in recent years, their present-day plumage appears more similar to that of adult birds

moult, as well as the more abundant food resources during the moulting period, may allow the birds to moult more feathers, resulting in a more extensive moult, with potential advantages for flight efficiency, thermoregulation and ornamentation.

How feather moult affects bird appearance and thereby sexual interactions is a largely overlooked aspect of avian biology. Unlike adults of most passerine species, which moult their whole plumage, juveniles of most Western-Palearctic passerines, stressed by time and limited by their overall inferior physiological condition and foraging competency[28–30], only partially moult

after fledging[9]. As a result, juveniles are often easily identified by their partial moult and nest-grown feathers, signalling youth and associated with low competitive ability[11]. A more extensive partial moult by juveniles may result in plumage that is more adult-like (Fig. 1c). A more extensive moult is likely to increase the attractiveness of males at a younger age and may positively affect bird fitness through sexual selection[31], though birds that underwent their post-juvenile moult and consequently have a more adult-like appearance and increased attractiveness may also be subject to more aggressive behaviour from competing adults, for

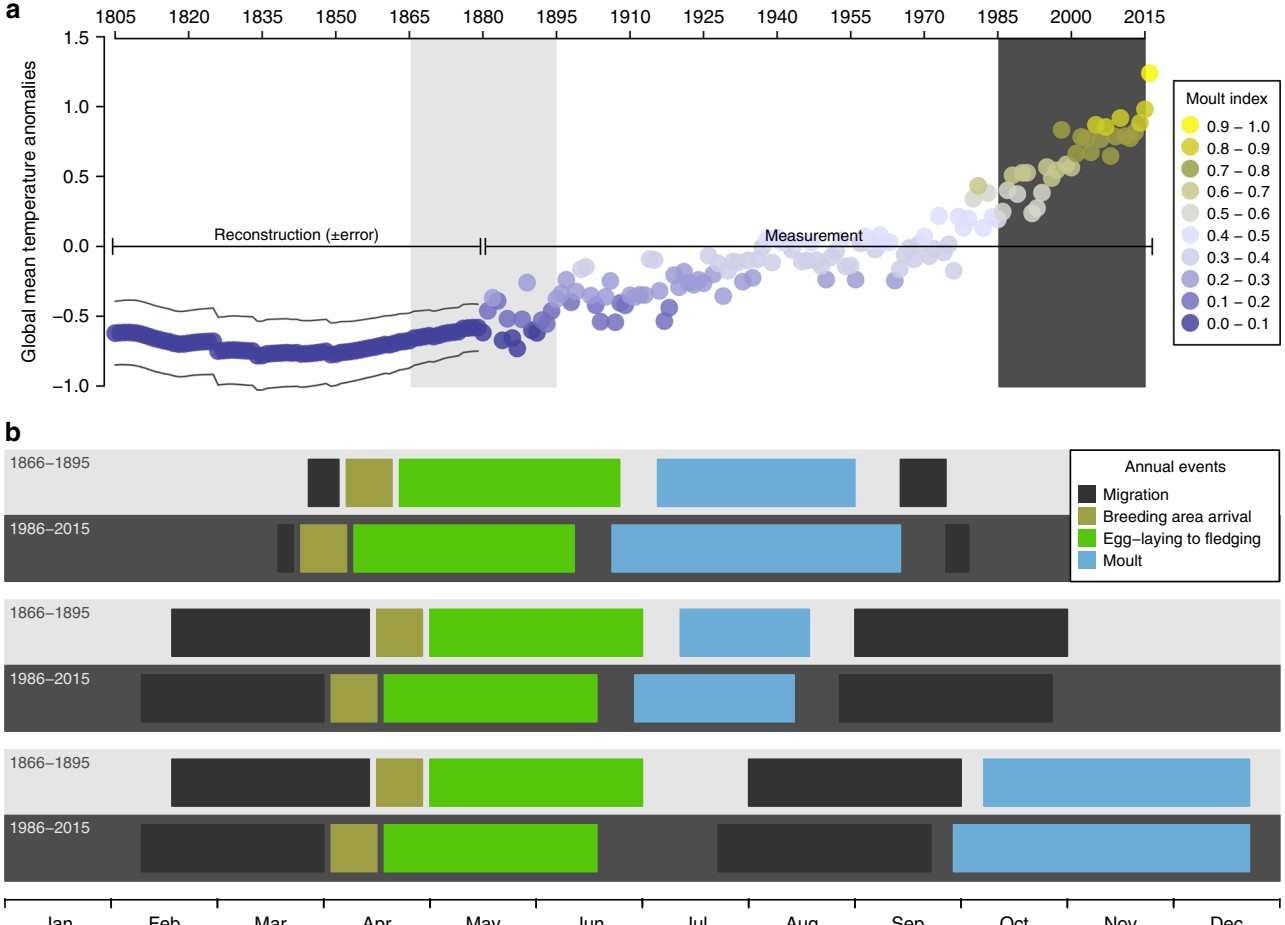

**Fig. 2** Passerine moult extent and phenological changes associated with global warming. **a** Global mean temperature anomalies (GMTA) trend since 1805 and bird feather moult extent. The moult index represents the relative extent of moult that was found in the study, averaged per year among all study species, such that the lowest relative extent of moult = 0 and the highest relative extent of moult = 1. Light grey and dark grey shaded areas represent two periods that are referred to in b. **b** Phenological changes in avian annual routine events over the last two centuries. Upper two rows (light grey = earlier years, dark grey = recent years): residents and short-distance migratory species – advancement of spring arrival and breeding, delayed autumn migration and shortened migration distance causing a longer moulting period and resulting in a more extensive moult. Middle two rows: Long-distance migratory species with pre-migration moult – advancement of spring arrival and breeding causing a longer moulting period, also resulting in a more extensive moult. Lower two rows: Long-distance migratory species with post-migration moulting period – it is suggested that advancement of the autumn migration combined with earlier arrival to the wintering areas and richer food resources in the northern Afrotropical Sahel zone after the summer rains allow these birds to moult more extensively. The changes in the timing of annual routine events are largely based on data from different published studies[3–5,15,22]

example during territorial disputes[12]. We note that, in general, females have higher nesting and post-fledging parental investment than males[32–34], likely exposing the females' plumage to higher mechanical strain that may lead to higher feather abrasion during the breeding period[35]. Moreover, the onset of the post-breeding moult, which follows the post-juvenile moult, is later among females than among males[36–39]. Thus, plumage of females that moulted extensively during their post-juvenile period may be more durable in the breeding period. Thereby, an extensive post-juvenile moult may positively affect bird fitness, also among females.

Males and females differ in several behavioural, physiological, ecological, morphological and chromatical characteristics: e.g., thermoregulation[16,17], migration distance[19] and colourfulness[40], which are predicted to interact differentially with climatic conditions. Our results demonstrate a stronger response of females than males in moult extent under climate warming in four species out of 10 sexually dichromatic species, and in no species males moulted significantly more extensively than females (Fig. 1b and Supplementary Table 3). In these four species, GMTA, sex and

their interaction affected moult extent. We note that the response of females to GMTA is stronger in highly sexual-dichromatic species as oppose to species in which males and females have similar ornamentation (Fig. 3). It is not clear why the response to GMTA is influenced by sexual-dichromatism, and we speculate that two factors may be involved: (1) feather durability and (2) ornamentation plumage cost. Feathers characterised by blackish and melanin-rich colour are more durable than non-melanic feathers[41,42], and hence, to overcome their larger feather wear, females may benefit from a more extensive moult compared with males, but only in species in which females have lighter and therefore less durable plumage than males. The other explanation is based on the costs and benefits of ornamented feathers of males. While these ornaments could provide mating advantages, they may also induce a cost, for example, if sex-biased mortality or a more aggressive behaviour occurs as a result of a more ornamented male plumage[12,43,44]. We note that our results regarding the differences between males and females in the response to GMTA in relation to their sexual-dichromatism level are based on a rather small sample size (10 species), and

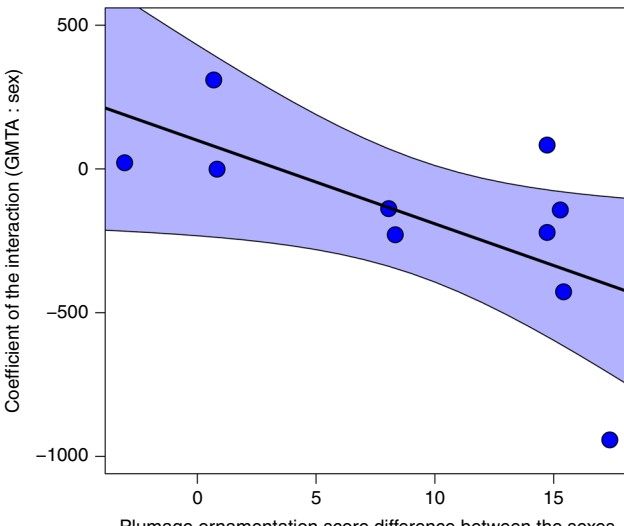

**Fig. 3** The relationship between plumage ornamentation differences between male and female birds. Plumage ornamentation score differences between male and female birds is calculated as ($male_{ornamentation}$ − $female_{ornamentation}$). We illustrate the interaction in the models that tested the effects of global mean temperature anomalies (GMTA) and bird sex on the area of moulted feathers in the post-juvenile moult in relation to plumage ornamentation score differences among sexual-dichromatic bird species tested in the study ($n = 10$). In species with a larger difference in plumage ornamentation between the sexes, the response of females to GMTA was high compared to that of males, as shown here by their overall lower coefficients of the GMTA and sex interaction, with male relative to female values. The plumage ornamentation score is provided by Dale et al.[40]. The method used for quantifying plumage colouration for male and female of each species includes red, green and blue (RGB) measurements in three dorsal (nape, crown, forehead) and three ventral (throat, upper breast, lower breast) body patches. For each plumage patch, RGB values for both sexes in all the ten species were collected

hence, more study is needed for substantiating this interesting relationship.

A positive effect of climate warming on the number of moulted feathers among passerines is likely a result of cumulative phenological changes across several annual routine events, including breeding and migration. During the ~200 year period examined in this study, GMTA increased by almost 2.0 °C. Current projections suggest an increase of another 1.5–4.5 °C by 2100[45,46]. It is still unclear to what extent the response of bird moult to climate warming is due to phenotypic plasticity, as opposed to genetic change[47–49]. Either way, given the predicted increasing climate warming over the next decades, we believe that post-juvenile moult will become even more extensive with time. This alteration from limited to extensive partial post-juvenile moult under a warming climate is predicted to transform physiological, behavioural, morphological and ecological features of avian life in measurable ways in a large number of bird species.

## Methods

**Data collection**. Data were obtained from bird skins stored in the collections of 10 natural history museums: (I) Natural History Museum (Tring; UK), (II) Museum National d'Histoire Naturelle (Paris, France), (III) National History Museum of Denmark (Copenhagen, Denmark), (IV) Museum für Naturkunde (Berlin, Germany), (V) Museo Nacional de Ciencias Naturales (Madrid, Spain), (VI) Naturhistoriska Riksmuseet (Stockholm, Sweden), (VII) Natural History Museum Vienna (Vienna, Austria), (VIII) Steinhardt Museum of Natural History, Tel-Aviv University (Tel-Aviv, Israel), (IX) National Natural History Collections, the Hebrew University of Jerusalem (Jerusalem, Israel) and (X) Naturhistorisches Museum (Basel, Switzerland). Additionally, we used moult data that was collected during bird ringing activities in Spain, Italy, Switzerland and Israel. We also used

published bird photos for species in which it was possible to identify moulted and non-moulted feathers from photos, e.g., from the Internet Bird Collection (IBC; https://www.hbw.com/ibc). The age of each bird was identified using unique published characteristics for each species[50,51]. Moreover, regularly, juvenile nest-grown feathers are poorer in texture, duller and characterised by higher abrasion level than that of feathers growing during the post-juvenile moult or by adults[51,52]. Using all these plumage characteristics, each individual was aged as either a first-year (post-juvenile) or an adult bird (always older than first-year). We identified post-juvenile individuals as those which had completed their partial feather renewal but had not yet begun their first complete adult moult. Each wing and tail feather of first-year individuals was classified as being moulted or non-moulted. Moult extent was determined by documenting the moult of the wing and tail feathers using a score of 0 (non-moulted feather) or 1 (moulted feather). A total of two feather tracts and 48 feathers were documented using this method for each individual, as follows: lesser-coverts (LC; proportion), median-coverts (MC; proportion), greater-coverts ($GC_{1-10}$), carpal-covert (CC), alula ($Al_{1-3}$), primary coverts ($PC_{1-9}$), primaries ($P_{1-10}$), secondaries ($S_{1-6}$), tertials ($T_{7-9}$) and rectrices ($R_{1-6}$). For two feather tracts with numerous small and difficult to distinguish feathers (lesser- and median-coverts) we only estimated the moulting proportion. Then, we calculated the species-specific mean area of each tested feather tract ($mm^2$) using photos that were taken in the field on live birds or on bird skins in the museums. We measured feather tract area using Adobe Photoshop version 6.0.1.a (Supplementary Table 7 and Supplementary Fig. 2). Then, based on the moult data, we calculated the area of the moulted plumage area in the wing and the tail of each individual.

**Study species**. The study includes Western-Palearctic species of the genuses *Turdus* (Turdidae), *Oenanthe* (Muscicapidae), *Monticola* (Muscicapidae), *Ficedula* (Muscicapidae) and *Lanius* (Laniidae), of which we measured more than 100 individuals each (Supplementary Table 1). The identification of moulted versus non-moulted feathers in these species is straightforward (Supplementary Fig. 3), permitting the generation of a reliable dataset. For each species, we collected data for all subspecies that are common in the Western-Palearctic region (*e.g.*, *niloticus*, *senator* and *badius* subspecies of *Lanius senator*; Supplementary Table 1). Since *Lanius excubitor* displays a markedly different moult strategy (including moult extent) between its northern subspecies (*excubitor*) and southern ones (*aucheri*, *elegans* and *algeriensis*; Supplementary Table 6), we considered this species in our analyses as two separate species: *Lanius* (*excubitor*) *excubitor* (Great Grey Shrike) and *Lanius* (*excubitor*) *aucheri* (Southern Grey Shrike), as has been done in several previous studies[53–55].

**Moult timing and location**. The birds were tested after the end of their post-juvenile moult and before their first post-breeding moult. We determined that moult has ended since we visually inspected each feather and could not trace signs that the moult is still undergoing by recording feathers that are shorter in size and feathers with remains of their protective peripheral sheath. In species that moult before autumn migration, examined individuals were between ~3 and ~12 months old (approximately during September–June). In species that moult after the autumn migration, in the over-wintering areas, examined individuals were between ~7 and ~12 months old (approximately during January–June). In these periods, no additional moult is undertaken and no feathers are replaced (in the study species, a pre-breeding moult is rare and very limited or does not exist). Thus, examining birds that were collected or examined in the field during these periods reliably represent the extent of the post-juvenile moult that occurred earlier, either near the breeding areas or, in the case of post-migration moult, in the over-wintering areas. We note that several studies suggested that the extent of feather moult is affected by latitude: southern species were found to moult more extensively than northern ones[9,30]. Due to the uncertainty involved with determining the geographic location of moult in birds that were collected throughout the year, often thousands of kilometers away from their natal, breeding and moulting areas, we could not include the latitude of the moulting area in our analyses. Each museum collection used in the study consists specimens of various sources that were collected in different localities. By using data from 10 bird collections and from live birds that were examined in several countries (Supplementary Table 2), it can be assumed that we created a largely randomly sampled dataset.

**Global mean temperature anomalies**. We used the global mean temperature anomalies (GMTA) as a proxy for global temperature in our analyses. GMTA data for the period 1880–2016 were obtained from the National Aeronautics and Space Administration (NASA), Goddard Institute for Space Studies (https://data.giss.nasa.gov/gistemp/). These anomalies data were calculated relative to the mean temperature of the 1951–1980 reference period. The reconstruction of GMTA data for the period 1801–1879 was obtained from published data that is based on glacier length fluctuations[56]. We note that GMTA, by its nature, is an index that cannot capture small-scale variation over local spatial scales and within short-time scales (e.g., between different months). Therefore, the ecological relevance of this index is limited when inferring different biological processes that take place over smaller spatial and temporal scales in which fine-scale variation is important.

**Statistical analysis**. In species that are sexually monomorphic (n = 9; Supplementary Table 1), for which the individual's sex could not be morphologically or chromatically determined, we used a generalised linear model (GLM; family = Gamma; $g(\mu_i) = \mu_i$; using R package 'lme4'[57]) to explore the effects of GMTA on the extent of moult. In sexually dichromatic species (n = 10) we used GLM as well to explore the effects of GMTA, sex and their interaction on the extent of moult and selected the best model based on the AICc[58]. Each species was tested separately and we selected a specific model only if it exhibited a ΔAICc > 2.00 compared to other models. For the sexually dichromatic species, we also examined the influence of the difference in plumage ornamentation between the sexes on the effect of GMTA on the extent of moult. In this analysis, we used plumage ornamentation score data that were published by Dale et al.[40]. We note that the available data period differ between the species that are included in the study (start year range: 1805–1864, mean = 1829; Supplementary Table 1), and hence model slope coefficient values should not be compared between species that differ in their starting year of data period. Since migration distance (km) and moult strategy (pre- or post-migration moult) are known as factors that may affect bird moulting properties[27], we tested their effects on the relationship between feather moult extent and GMTA. The migration distance was calculated as the distance between the mid-breeding and the mid-wintering areas of each species (BirdLife International and NatureServe 2014[59]). Because species traits are known to be phylogenetically conserved, and thus data from closely related species are not statistically independent, we repeated the analysis following the independent contrasts method, which identifies evolutionarily independent comparisons[60]. To account for phylogenetic non-independence, we conducted all analyses using PGLS regression[61]. We examined the strength of phylogenetic non-independence using the maximum likelihood value of the scaling parameter Pagel's $\lambda$[62] implemented in the R package 'caper'[63]. Pagel's $\lambda$ is a multiplier of the off-diagonal elements of the variance-covariance matrix, which provides the best fit of the Brownian motion model to the tip data, and ranges between zero (no phylogenetic signal) and one (phylogenetic signal that depends on branch lengths as in analysis of phylogenetic-independent contrasts). We then corrected for the effects of shared ancestry using the maximum likelihood value of $\lambda$. The phylogenetic tree (Supplementary Fig. 4) was obtained from an analysis of global bird diversity[21] using 10,000 trees that were generated from BirdTree.org[64]. The consensus tree was built using BEAST version 1.8.4. Analyses (two-tailed, critical $\alpha$ = 0.05) were performed using R (version 3.5.3; R Development Core Team 2016).

## Data availability

The data that support the findings of this study are available from the corresponding authors upon reasonable request. The parameters that were used in all our analyses and figures are provided as a Source Data file.

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

## Acknowledgements

This research received support from the Synthesys Project which is financed by the European Community Research Infrastructure Action under the FP7 project (call 3) at the Museum National d'Histoire Naturelle (FR-TAF-5571), National History Museum of Denmark (DK-TAF-6443), Museum für Naturkunde (MfN; DE-TAF-6398), Museo Nacional de Ciencias Naturales (MNCN; ES-TAF-6445), Naturhistoriska Riksmuseet (NRM; SE-TAF-6442) and Natural History Museum Vienna (NHMW; AT-TAF-6444). We would like to thank H.V. Grouw and M. Adams from the Natural History Museum (Tring; UK), J. Fuchs and V. Bouetel from the Museum National d'Histoire Naturelle (Paris, France), K. Thorup, J.B. Kristensen and N. Manniche from the National History Museum of Denmark (Copenhagen, Denmark), S. Frahnert, P. Eckhoff and M. Voß from the Museum für Naturkunde (Berlin, Germany), J.B. Rodriguez and J. Cabarga from the Museo Nacional de Ciencias Naturales (Madrid, Spain), U. Johansson and I. Bisang from the Naturhistoriska Riksmuseet (Stockholm, Sweden), A. Gamauf and A. Hille from the Natural History Museum Vienna (Vienna, Austria), D. Berkowic and A. Belmaker from the Steinhardt Museum of Natural History (Tel-Aviv, Israel), R. Efrat from the National Natural History Collections, the Hebrew University of Jerusalem (Jerusalem, Israel) and R. Winkler from the Naturhistorisches Museum (Basel, Switzerland). Thanks also to R. Efrat and Y. Belmaker for their assistance with the statistical analysis and to P. Dougalis for drawing the bird illustrations in Figure 1.

## Author contributions

Y.K. and N.S. conceived the study. Y.K. collected the data in natural history collections, conducted statistical analyses and prepared figures with the assistance of N.S. and Y.V. Y.K. wrote the initial draft of the manuscript, which was subsequently revised by N.S. and Y.V.

## Additional information

**Competing interests:** The authors declare no competing interests.

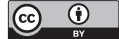

7