## [Peer Review File · Nature Communications]

Reviewers' comments:

Reviewer #1 (Remarks to the Author):

The authors have examined an outstanding number of birds to study how moulting patterns in several species have changed during the last two centuries. Moulting is a key process in the annual cycle of birds, but surprisingly this phenomenon receives little attention. Thus, I fully agree with the authors' statement (L115): "How feather moult affects bird appearance and thereby sexual interactions is a largely overlooked aspect of avian biology". For this reason, I began to read very excited this ms because it looked really promising and an important contribution. Unfortunately, in spite of the fact that the study objectives are excellent, novel and ambitious, I have serious concerns about the validity of the main conclusions of the authors. My concerns arise from the insufficient methodological information to assess the validity of data, as well as from some potential artefacts in the results. Such concerns are summarized in the next main points:

1) You mixed information from museum skins and alive birds. I guess moulting was assessed by different observers in each case. How did this fact affect moulting scores? To what extent both information sources are comparable? You would need to provide the sample size coming from each source too.

2) You do not provide any information about the date and site of collection of the examined individuals. I hope you used only juveniles collected during the autumn, when postjuvenile moult have been completed. But this is just an assumption. Based on your ms, I may alternatively interpret your results as an artefact due to a tendency to collect individuals later in the autumn in the XX century compared to the XIX. If they have been collected later, they can have a more extensive postjuvenile moult. Actually, how did you know that postjuvenile moult was ended in the museum skins? Collection site may induce a systematic bias as well. If you have more individuals from northern populations in the last decades, you may observe a more extensive moult. Date and site are important confounding factors, which have been omitted in the ms (but see L299-302).

3) In spite of the outstanding number of examined birds, samples are relatively small for a such long period of two centuries. For most species you have <1 indiv/yr. As previously, you do not provide any information about the temporal distribution of your samples. I can hardly imagine that birds were homogeneously collected along the full study period. I guess most of them were concentrated in some or few years. I would like to know the effect of such (possible) patching along time of your samples in the estimated temporal trends. Moreover, the first available year varied between 1805 and 1864. Almost 60 years of difference in the onset year of your moulting time-series may have a

serious effect on the estimated trends. Note that your title is only valid for 6 out 19 (ie minority) studied species...

4) The positive correlation between moulting and temperatures looks just a result of temporal collinearity of both variables. Frankly, I do not trust in the main result of the study. I would like to see the regression models including year as an explanatory variable as well. Note that a mere demonstration of a temporal trend in moult extend would be already a very relevant result. However, as pointed above, this trend can be also an artefact due to several potential systematic biases in your dataset.

5) I do not understand why you used GMTA, if you are studying a process happening mostly in Europe and the Middle East. Once again, I just can make assumptions because no info about the provenance of studied birds is provided. In any case, I would suggest to use a more specific climate, eg temp in Europe.

Minor, but not less important, comments:

L24: maintaining plumage utility

L27: remove ornamental

L31: Could you rewrite this sentence to be more informative? Could you specify eg how temp influence moult?

L35: remove an

L36-7: Could you be more informative? You may explain briefly what those studies found

L50: This claim is false. Looking at table S2 I can see only 5 species in which the interaction sex*GMTA was significant. Moreover, you simply show the significance of this interaction. This is not informative to know whether females or males had the strongest response to temp.

L53-4: This claim is wrong! ...meaning that females replace more feathers than males under warmer conditions.

L83: remove "and indication of reduced migration distance".

L87-8: Nice hypothesis, but note that it would be only valid for SDM, which are mainly Turdus in your dataset.

L119-121: This claim is right, but juveniles are still undergoing a partial moult and thus are distinguishable from adults.

L122-5: What about juvenile females? They are changing more the extent of their moult and consequently they have more adult-like appearance than males. What can be the benefit for them?

L132-9: Good hypothesis. Another non-excluding one could be that females moult later than males. This happens in postbreeding moult of adults (>1 years old), but I am not sure whether it also happens in postjuvenile moult of juveniles. If females moult later, they would be more time constrained to do it before migration. However, if fledgling advances and migration delays, females would be the most benefited of this enlarged period in between for moulting.

L149: As I stated previously, juvenile moult is still partial. Thus, you cannot talk about complete juvenile moult.

L271: identify

L277: right or left wing?

L281-2: Are those proportions between 0 and 1? Thus, are you giving the same weight to all LC or MC than a single feather of the rest of types? Frankly, this moulting score is not fully convincing... While moulted/unmoulted is a binary state, all feathers are not equivalent in terms of cost and extent.

L286: remove common. Eg *O. lugens* is not a common Palearctic bird...

L289: Remove "producing a 289 dataset of 101-491 individuals per species (mean 211 individuals; table S1)."

L290: straightforward

L291: generation

L310: I think you need to use generalized linear models because the moult score is a Poisson variable.

L312: I think model selection in this context is totally unnecessary. You have only a couple of explanatory variables (temp and sex), both with well-defined a priori hypothesis of their effect. Actually, you want to test both variables. Thus, I would carry out full models in all dichromatic species (ie temp*sex).

Fig1. Circle insets are misleading in their current position. They should be placed in the boxed inset. The moult pattern under warm temperatures shown in the right circle of panel A looks really weird. Postjuvenile moult always involves coverts and inner secondaries. Pictures of C are really nice, but the difference between juvenile and adult feathers is excessively exaggerated. Unfortunately, differences between juvenile and adult feathers are subtler than that depicted.

Table S1:

- Is *Lanius isabellinus* referring to the former complex of subspp (currently split into: *L. isabellinus*, *L. phoenicuroides*, *L. speculigerus*) or to the new species (former subspp *L. i. isabellinus*, *L. i. tsaidamensis*, and *L. i. arenarius*).
- Migration distance seem a bit excessive for some spp, such as *O. deserti*, *M. solitarius* or *T. merula*. Note, for instance that *M. solitarius* is migrant only in the Caucasus region. *T. merula* is rather a partial migrant, ie only part of individuals moves.
- Sexual dichromatism is hard to distinguish in juveniles of *O. hispanica*, *F. hypoleuca* and *L. senator*.

Table S3: In a simple t-test, the df would be 366. Maybe you need to provide more details about this analysis. Moreover, see my comment above about non-normality of moult score.

In the reporting summary, you state that the study did not involve field work. This disagrees with the statement of L270 as well as with your statement about author contribution.

Reviewer #2 (Remarks to the Author):

This is an interesting and timely study that utilizes natural history collections (NHCs) to look at changes in molting in juvenile birds over the past 211 years and whether this can be linked to climate change. This paper helps to advance our knowledge about how species are responding to climate change.

However, I do have several concerns. Neither the methodology nor the analysis included enough details. Consequently, I have a hard time understanding the underlying data structure and evaluating whether the appropriate statistical technique was used. I also have some concerns about the inferences made by the authors based on the results they found.

Methodology

> What were the localities associated with these specimens? A map would be very helpful. It was difficult to get a sense of the extent of geographic variation present in the data.

>What are the dates of collections of these specimens? How much did they vary within a year? This could be problematic based on the potential mechanisms related to phenology and timing.

> Is the locality associated with the specimen the same as the geographic moult location (line 299)?

> Why was field data added to the collections, what years and localities?

Influence of temperature

> The authors didn't provide enough information about the role of temperature in this molting process, particularly at the scale of this study. At what spatial and temporal scale do temperature conditions influence molting? Do temperatures at the molting areas affect the extent of molting?

> The authors discuss two potential mechanisms by which temperature can influence molting- via time or resources. Are there any direct effects of temperature (i.e. physiological) on the process of molting, for example mediated through changes in developmental rate?

> Why were Global Mean Temperature anomalies used? How were the anomalies calculated? What about geographical variation in temperature change? Are the authors arguing that geographical variation in temperature change is not important i.e. a cold year is a cold year no matter where you are? There needs to be more clarity about the ecological relevance of this variable and the potential limitations of using it.

> There is almost certainly greater uncertainty surrounding the temperature measurements in the first time period. Are the results sensitive to the removal of this dataset?

Analysis

> Were the assumptions of linear (parametric) models evaluated and met?

> By modelling each species individually, the authors are assuming each species is responding differently to temperature. This could overstate the degree of interspecific variation and influence the estimate of central tendency. Did the authors consider using mixed-effects models? Also concerning is that the uncertainty surrounding the relationship between GMTA and molting extent for each species will vary given the differences in sample size and number of years sampled across species. Was this taken into account?

> How was the general trend and its CI evaluated? What is the effect size?

> Why was the effect of migration distance and moult strategy tested?

> Did a PGLS model better fit the data than a GLS? To me it's not clear whether a PGLS model was needed. Why was Brownian motion the only hypothesis considered?

Results

- > Rather than reporting a range of t-values across species (e.g. line 44), authors could tally up the number of species with a significant fit.
- > Where are the supporting results (i.e. species-level) for the temporal trend analysis?

Data presentation.

Figure 1.

- > Authors need to explain how to interpret the temperature axis.
- > In the inset figures, the authors have not shown the underlying data supporting the general trend.
- > Caption for 1B is confusing because the authors show relationship with temperature AND across time but only mention the temporal trend in the caption.

Figure 2.

- > I find this Figure confusing. Do parts B-D directly relate to the references cited in the caption? Or do the authors infer temporal changes in the annual events based on the results from the previous studies? This balance needs to be better articulated. Are there estimates of the time period of moulting available for each year? If so, why wasn't this directly related to changes in molting extent?
- > How was the moult index calculated? Proportion of which trend?

Conclusions

The authors conclusions about the differences between sexes are overstated (e.g. 15-18, lines 49-50, line 52-54, line 128-129). The authors only found a difference between sexes for 5/10 species. This is only half the time. Also, finding a relationship between molting extent and GMTA does not necessarily mean that there's a trend through time, as the authors infer in lines 52-54.

The conclusions about the role of temperature is also overstated in some places (e.g. line 41 and title of paper). I don't think the results following the analyses conducted in this paper demonstrate that climate change has changed bird appearance, as the title suggests. The authors showed directional change in proportion of feathers molted over time and showed that there were correlations with inter-annual variation in temperature (again assuming I understand what GMTA is capturing based on the details provided). Specifically, the authors showed that warmer years lead to more extensive

molting (assuming that positive values of GMTA =warmer years, see comment about Figure 1) but I'm not convinced they showed that directional changes in temperature over time led to these changes in bird appearance.

Minor comments

>line 8: The first I read this through I wasn't sure what the authors meant by 'individual appearance'. Because this journal has a general audience, it would help to clarify what you mean here. For example it could also be interpreted to mean their first appearance within a season.

> What plumage characteristics were used to determine the age of the bird? Is there any uncertainty with identifying first year juveniles? Are adults any bird older than first-year?

>I think it would be helpful to have a diagram of the anatomy of the feathers showing which parts were assessed for molting extent.

> several typos (e.g., lines 29, 34)

> In reading the introduction, my question immediately was- what are the implications if juveniles look more like an adult? This is addressed in lines 114-125. I would suggest moving this section to the introduction as it helps to better set up the study.

> line 34- what differences about the sexes would lead to differences in thermoregulation and behaviour (e.g., size? plumage?)

> Table S2- The coefficient of the interaction term for those species where there was a significant interaction should be included.

Reviewer #3 (Remarks to the Author):

This is a well-executed manuscript showing that the extent of the post-juvenile (preformative) molt has become more extensive over the past 200+ years among 19 European bird taxa. Text could stand a bit of copy-editing (e.g., Summary paragraph, 7th and 10th lines) but length is appropriate, methods and statistical procedures seem sound, sample sizes are excellent, figures are well-designed and informative, and supplementary materials are adequate. Findings are novel and will be of interest to students of bird molt and those tracking the effects of global climate change.

I have a few rather minor specific comments:

The term "juvenile" is widely used to refer to a bird in juvenile plumage. It then undergoes a preformative (or "post-juvenile") molt during the first 1-6 months of life, after which it is in formative (post-juvenile) plumage. A sentence summarizing this and these terms would be useful early in Main text to put the study in context (replace 3rd-4th sentences). Throughout ms. carefully construct wording accordingly. E.g., "...causes juveniles to look more like adults..." (Summary paragraph, line 11) is incorrect as it is the resulting formative plumage that will more-closely resemble that of adults (in some species) if the preformative molt is more extensive. "...juvenile moult" (e.g, Fig. 1 caption) should be replaced by post-juvenile or preformative moult, etc.

Main text, 4th sentence: Point out that the preformative molt can vary widely in extent on both an individual and taxa-specific context (as related to available time and resources)

Main text

1st paragraph, last line: Point out that these two previous studies examined timing of molt. This study is the first to examine extent to my knowledge.

2nd paragraph: Establish here that the study species breed in (northern?) Europe and show various migration strategies. [Since affects of global climate change on bird molt will vary widely around the globe, it is important to put this study into a boreal-centric context at the outset.]

3rd paragraph, 1st line: Recent evidence indicates that most post-breeding and post-juvenile molting does not occur "at the breeding grounds" - better to say "near" the breeding grounds or within the breeding range of the species.

Three taxa show no migration and are thus "resident" (Table S1). It could be worthwhile to separate these. Related, it would seem earlier breeding (II) would occur in all species, not just the migrant species.

5th paragraph - Females of course also differ in their breeding biology, in many species resulting in later molt after more-extensive nesting and post-fledging responsibilities than males.

The argument that the sex-specific difference is related to feather melanin does not hold on its own. Authors would first need to demonstrate that males had molted fewer feathers than females before climate-change effects were incurred. I know of no study that demonstrates this. Alternatively, there could be an effect if more melanin is produced later during the post-juvenile molt in males. This effect (termed "molt-plumage interaction" - Pyle 2013a) is seen in North America, particularly among long-distance migrants such as orioles and *Pheucticus grosbeaks*, whereby feathers replaced later during the molt show more adult (definitive) characters as the timing of molt advances, including the production of melanin. Do all five species showing this sex difference have sex-specific plumage involving melanin and, conversely, do the five species not showing the sex difference lack extensive feather melanin? This would be good to know. In general I think the melanin connection can be mentioned but not emphasized, and that other explanations related to male-female differences in behavior, e.g., timing of molt related to breeding, should be proposed.

Pyle, P. 2013a. Dark-faced Common Murres off California in fall and winter. *Western Birds* 44:250-261.

Sixth paragraph: Molt extent has been proposed as being very plastic due to extensive variation both within species and between closely related species that, e.g., demonstrate differing migration strategies (Pyle 2013b).

Pyle, P. 2013b. Evolutionary implications of synapomorphic wing-molt sequences among falcons (Falconidae) and parrots (Psittaciformes). *Condor* 115:593-602.

Final two lines. Whether or not the preformative molt can vary to complete is not a linear function in passerines and varies phylogenetically. I doubt that most of the study species (e.g., any of the Turdidae) will shift to having a complete preformative molt (i.e., including all remiges) as a response to climate change (Turdidae are relatively fixed in molting variable numbers of upperwing coverts but few if any primaries). Recast this simply in terms of more feathers replaced with time. Further, complete preformative molts being "a common pattern in tropical species" is true to a minor but not a major extent, at least in Neotropical passerines (Johnson and Wolfe 2018), and it would appear to have more to do with solar exposure among species than anything to do with temperature. Thus, the implicit connection between global warming and molt extent with this statement should not be suggested.

Johnson, E. I., and J. D. Wolfe (2018). Molt in Neotropical birds: Life history and aging criteria. *Studies in Avian Biology* 51.

Methods -

Second paragraph - "...northern species molt more extensively than southern species..." I do not believe this to be true in any context (e.g., see above re tropical vs. boreal species) but even within temperate zones the opposite seems more the case, at least in North America. Remove this as it has little bearing on this ms.

Hope this helps

Peter Pyle

ppyle@birdpop.org

Comments by Reviewer #1:

The authors have examined an outstanding number of birds to study how moulting patterns in several species have changed during the last two centuries. Moulting is a key process in the annual cycle of birds, but surprisingly this phenomenon receives little attention. Thus, I fully agree with the authors' statement (L115): "How feather moult affects bird appearance and thereby sexual interactions is a largely overlooked aspect of avian biology". For this reason, I began to read very excited this MS because it looked really promising and an important contribution. Unfortunately, in spite of the fact that the study objectives are excellent, novel and ambitious, I have serious concerns about the validity of the main conclusions of the authors. My concerns arise from the insufficient methodological information to assess the validity of data, as well as from some potential artefacts in the results. Such concerns are summarized in the next main points:

Our response: We hope that we now provide sufficient information about our methodology and clarify the results in the updated version of the manuscript. Please find below our responses to your specific comments.

Comment 1: 1) You mixed information from museum skins and alive birds. I guess moulting was assessed by different observers in each case. How did this fact affect moulting scores? To what extent both information sources are comparable? You would need to provide the sample size coming from each source too.

Our response: 95.2% of data (from museum skins and live birds) were collected by Yosef Kiat (the first author). All the rest of the data were collected by Dr. Raffael Winkler (author of the book "Moult and ageing of European passerines", 1994). Both researchers are highly experienced in recording feather moult data with tens of thousands birds recorded by each of them. We further note that registering feather moult by different observers is not expected to affect our data because we deliberately chose to study species in which the differences between moulted and non-moulted juvenile feathers are easy to identify (for example, see Figure S3 that we added to the supporting material of the updated manuscript). Following this comment we added Table S2 which provides the sample sizes of the data from each source.

Comment 2: 2) You do not provide any information about the date and site of collection of the examined individuals. I hope you used only juveniles collected during the autumn,

when postjuvenile moult have been completed. But this is just an assumption. Based on your MS, I may alternatively interpret your results as an artefact due to a tendency to collect individuals later in the autumn in the XX century compared to the XIX. If they have been collected later, they can have a more extensive postjuvenile moult. Actually, how did you know that postjuvenile moult was ended in the museum skins? Collection site may induce a systematic bias as well. If you have more individuals from northern populations in the last decades, you may observe a more extensive moult. Date and site are important confounding factors, which have been omitted in the ms (but see L299-302).

Our response: Bird moult details: The birds were examined after the end of their postjuvenile moult and before their first post-breeding moult. In species that moult before autumn migration, the individuals were examined when they were ~3 to ~12 months old (approximately September to June). In species that moult later, after the autumn migration, in the over-wintering areas, the individuals were examined then they were ~7 to ~12 months old (approximately January to June). In these periods, no additional moult is undertaken and no feathers are replaced (in the study species, a pre-breeding moult is rare and very limited or does not exist). Thus, examining birds that were collected or tested during these periods reliably represent the extent of the post-juvenile moult that occurred earlier, either near the breeding areas or, in the case of post-migration moult, in the over-wintering areas. We added this information to the Methods (lines 420-431).

We determined that moult has ended since we visually inspected each feather and could not trace signs that the moult is still undergoing by recording feathers that are shorter in size and feathers with remains of their protective peripheral sheath. This information was also added to the Methods (lines 421-424).

The location of feather collection: we note that in most cases the birds likely moved between their moulting site and the collection site and in some cases, especially earlier in the study period, the location of bird collection was not properly documented. Since each of the ten bird collections combined specimens that were obtained by many collectors over a wide geographic area, and since live birds were measured in several different countries (and, as we explain above, many of which likely moulted in other localities and moved to the site where they have been measured), our data were largely randomly sampled and well represents the geographic extent of the Western Palearctic region. Following this comment, we added this information to the Methods section (lines 436-439).

Comment 3: 3) In spite of the outstanding number of examined birds, samples are relatively small for a such long period of two centuries. For most species you have <1 ndvi/yr. As previously, you do not provide any information about the temporal distribution of your samples. I can hardly imagine that birds were homogeneously collected along the full study period. I guess most of them were concentrated in some or few years. I would like to know the effect of such (possible) patching along time of your samples in the estimated temporal trends. Moreover, the first available year varied between 1805 and 1864. Almost 60 years of difference in the onset year of your moulting time-series may have a serious effect on the estimated trends. Note that your title is only valid for 6 out 19 (i.e. minority) studied species...

Our response: (1) The information about the temporal distribution of our sampling: Following this comment, we added a histogram of the temporal distribution of the moult data (see Fig. S1). (2) The effect of patching along time of our samples in the temporal trend: We performed an analysis that considered the weight of the data based on its percentage in the total number of data points. The weights were based on 40-year periods to reflect changes in sampling during the >200 years of the study. These results (see Table 1 below) are very similar to the results that are reported in the updated manuscript. (3) The onset of data sampling: we note that excluding data from earlier years is a wrong practice and data from all available sources should be included in the analysis, provided that the range of data available for each species is reported, as has been done in Table S1 of the updated manuscript. Moreover, our results are not affected by differences between the sampling periods of the different species, as evident from our findings that do not relate to differences in the duration of data sampling of the different species. Since the study spans over a period of 212 years (1805-2016), the title is justified. We will be happy to consider alternative suggestions for the title.

Table 1. An analysis that considered the weight of each year based on the amount of data collected during 40-year periods.

		r^2_{model}	Coefficient	GMTA	
				t	P
Oenanthe oenanthe	~ GMTA	0.19	118.43 ± 20.62	5.74	< 0.001
Oenanthe deserti	~ GMTA	0.20	221.56 ± 32.59	6.80	< 0.001
Oenanthe hispanica	~ GMTA	0.01	21.97 ± 14.61	1.50	0.13
Oenanthe pleschanka	~ GMTA	0.04	67.09 ± 24.48	2.74	< 0.01
Oenanthe lugens	~ GMTA	0.10	113.82 ± 34.94	3.26	< 0.01
Oenanthe leucopyga	~ GMTA	0.18	203.60 ± 49.31	4.13	< 0.001
Monticola solitarius	~ GMTA	0.10	209.50 ± 62.18	3.37	< 0.01
Turdus philomelos	~ GMTA	0.05	55.15 ± 12.19	4.52	< 0.001
Turdus iliacus	~ GMTA	0.21	118.34 ± 13.26	8.93	< 0.001
Turdus viscivorus	~ GMTA	0.01	-56.26 ± 39.07	-1.44	0.15
Turdus pilaris	~ GMTA	0.13	139.21 ± 19.45	7.16	< 0.001
Turdus merula	~ GMTA	0.01	1540.70 ± 237.00	6.50	< 0.001
Turdus torquatus	~ GMTA	0.01	33.88 ± 22.21	1.53	0.13
Ficedula hypoleuca	~ GMTA	0.00	-15.57 ± 26.49	-0.59	0.56
Ficedula albicollis	~ GMTA	0.20	661.69 ± 109.79	6.03	< 0.001
Lanius isabellinus	~ GMTA	0.12	874.20 ± 211.70	4.13	< 0.001
Lanius senator	~ GMTA	0.26	716.78 ± 93.55	7.66	< 0.001
Lanius e. excubitor	~ GMTA	0.09	241.56 ± 49.58	4.87	< 0.001
Lanius e. aucheri	~ GMTA	0.06	522.30 ± 213.10	2.45	0.02

Comment 4: 4) The positive correlation between moulting and temperatures looks just a result of temporal collinearity of both variables. Frankly, I do not trust in the main result of the study. I would like to see the regression models including year as an explanatory variable as well. Note that a mere demonstration of a temporal trend in moult extend would be already a very relevant result. However, as pointed above, this trend can be also an artefact due to several potential systematic biases in your dataset.

Our response: Due to collinearity of GMTA and year (the correlation between them is $r = 0.92$, $P < 0.001$), we could not include these two factors together in the same models. Following this comment we added a table in the supplementary material that summarizes the results of models in which we included the year instead of the GMTA (Table S4), in addition to a table in which GMTA was the explanatory variable (Table S3). To decide whether to include the year or GMTA, we used model selection procedure based on AIC scores (the selection is based on lower AIC value). We furthermore ran two GLMM models with species as a random factor and found that a model that included GMTA as an explanatory variable had an AIC value of 60160.5 (GMTA: Coefficient = 165.38 ± 6.63 , $t = 24.94$, $P < 0.001$) and a model that included the year as an explanatory variable had an AIC value of 60195.5 (year: Coefficient = 1.46 ± 0.13 , $t = 11.48$, $P < 0.001$). $\Delta AIC = 35.0$ strongly supports the inclusion of GMTA rather than the year as an explanatory variable in the analysis.

Comment 5: 5) I do not understand why you used GMTA, if you are studying a process happening mostly in Europe and the Middle East. Once again, I just can make assumptions because no info about the provenance of studied birds is provided. In any case, I would suggest to use a more specific climate, eg temp in Europe.

Our response: There are several reasons for the use of GMTA rather than any other, more local, index (e.g., temperature in Europe). The breeding and moulting distribution of all our bird species are extending over a much larger area than the area covered by any local index. For example, *Oenanthe pleschanka* moults in Europe and Asia, and *Turdus merula* moults in Europe, Asia and North Africa. Feather moult of passerines takes place in the breeding areas at northern latitudes, but, in some species also in the wintering areas, south of the Sahara. We furthermore note that moult extent is influenced by several annual cycle events (e.g., migration) that take place in different regions, as has been reported in numerous publications (for example, Jenni & Winkler (1994, Moults and ageing of European passerines), Barta et al. (2008, Philosophical Transactions of the Royal Society of London B: Biological Sciences), Kiat et al. (2018, Biological Reviews)) and as discussed throughout the manuscript (for example lines 460-462). Another reason for using GMTA instead of a local index is the lack of a reliable reconstruction data of local indexes for the period preceding 1880. Furthermore, there is high correlation between all relevant indexes we considered so that using a different index is not expected

to considerably affect our results (the mean Pearson correlation coefficient is 0.91 in these correlations; see Table 2 below).

Table 2. Pearson correlation coefficients between several temperature anomalies indexes (P-value < 0.001, for all).

GMTA used in this study													
Mann et al. (2009), Northern Hemisphere	0.874	0.977	0.974	0.903	0.934	0.895							
NASA: Northern Hemisphere	-	0.868	0.874	0.857	0.816	0.845							
NASA: Southern Hemisphere		-	0.908	0.983	0.896	0.987							
NASA: Southern Hemisphere			-	0.939	0.947	0.890							
Berkeley Earth: Global				-	0.950	0.971							
Berkeley Earth: Southern Hemisphere					-	0.907							
Berkeley Earth: Northern Hemisphere						-							

Comment 6: L24: maintaining plumage utility

Our response: Corrected.

Comment 7: L27: remove ornamental

Our response: Corrected.

Comment 8: L31: Could you rewrite this sentence to be more informative? Could you specify eg how temp influence moult?

Our response: The information was added (lines 44-46).

Comment 9: L35: remove an

Our response: Corrected.

Comment 10: L36-7: Could you be more informative? You may explain briefly what those studies found.

Our response: In both studies the moult advanced under climate change; we added this information to the manuscript (lines 49-52).

Comment 11: L50: This claim is false. Looking at table S2 I can see only 5 species in which the interaction sex*GMTA was significant. Moreover, you simply show the significance of this interaction. This is not informative to know whether females or males had the strongest response to temp.

Our response: In line 50 (now line 66) we wrote explicitly that this result is valid only for five species. Following this comment we added the word "five" also in the next sentence to emphasize it again and revised this part. Following comment #59, we also added the values of the coefficients of the statistical models to the supplementary material (Tables S3 and S4). Additionally, Figure 1B is very informative regarding the responses of males and females to GMTA.

Comment 12: L53-4: This claim is wrong! ...meaning that females replace more feathers than males under warmer conditions.

Our response: Thank you. The text was corrected (lines 68-69).

Comment 13: L83: remove “and indication of reduced migration distance”.

Our response: Please note that Visser et al. (2009; Climate change leads to decreasing bird migration distances. *Global Change Biology* 15(8):1859-1865) reported about the reduced migration distance in 12 passerines species and also Van Vliet et al. (2009; Changes in migration behaviour of Blackbirds *Turdus merula* from the Netherland. *Bird*

Study 56(2):276-281) demonstrated reduced migration distance by a short-distance migratory species (*Turdus merula*).

Comment 14: L87-8: Nice hypothesis, but note that it would be only valid for SDM, which are mainly *Turdus* in your dataset.

Our response: This hypothesis is valid also for *Lanius excubitor*, *Monticola solitarius*, *Oenanthe deserti*, and long-distance migratory species that moult before migration (for example, *Oenanthe pleschanka* and *Oenanthe hispanica*), as we explicitly mention in the next sentence: "This extension of the moult period could be advantageous to birds that moult before migration (short- or long-distance migrants)". Following this comment the sentence was revised (lines 104-105).

Comment 15: L119-121: This claim is right, but juveniles are still undergoing a partial moult and thus are distinguishable from adults.

Our response: Following this comment we added the word "partial" to emphasize that moult is still partial, although more extensive (line 140).

Comment 16: L122-5: What about juvenile females? They are changing more the extent of their moult and consequently they have more adult-like appearance than males. What can be the benefit for them?

Our response: Thank you for pointing this out. The benefit to females could be better resistance to wear and tear. Females are generally more involved in nest-related activities such as incubation, and therefore their feathers are subjected to mechanical strain in the nest which can lead to higher feather abrasion during the breeding period. Moreover, the onset of the post-breeding moult takes place later in females than in males, presumably due to their extended parental care. Consequently, a more extensive female moult likely increases the durability of feathers in females, thereby positively affecting bird fitness. We added this information to the end of the paragraph (lines 145-151).

Comment 17: L132-9: Good hypothesis. Another non-excluding one could be that females moult later than males. This happens in postbreeding moult of adults (>1 years old), but I am not sure whether it also happens in postjuvenile moult of juveniles. If females moult later, they would be more time constrained to do it before migration.

However, if fledgling advances and migration delays, females would be the most benefited of this enlarged period in between for moulting.

Our response: Thanks for the constructive comment. There is currently no support in the literature regarding sex-dependent differences in the timing of post-juvenile moult onset and we also could not find any evidence for that in our data. However, we added to the updated manuscript information about the difference in the post-breeding moult. See also our response to the previous comment (#16).

Comment 18: L149: As I stated previously, juvenile moult is still partial. Thus, you cannot talk about complete juvenile moult.

Our response: We removed any mentioning of complete juvenile moult from the updated manuscript and revised this sentence.

Comment 19: L271: identify

Our response: Corrected.

Comment 20: L277: right or left wing?

Our response: We tested regularly the right wing, but sometimes the right wing was not available and then we tested the left wing. Whether we examined the right or the left wing is not expected to affect our results since, usually, moult is symmetrically on both wings and if not, there should not be any asymmetrical tendency to for more advanced moult in one of the wings (Jenni & Winkler, 1994).

Comment 21: L281-2: Are those proportions between 0 and 1? Thus, are you giving the same weight to all LC or MC than a single feather of the rest of types? Frankly, this moulting score is not fully convincing... While moulted/unmoulted is a binary state, all feathers are not equivalent in terms of cost and extent.

Our response: Thank you for this thoughtful comment. Following this comment we completely changed the way we considered the moult data in our analyses and we now use the feather cover area instead of using the binary (moulted / non-moulted) index of the number of moulted feathers we previously used. The use of feather cover area likely better represents the extent of moulted feather area which is directly related to the cost of feather production and the appearance of the birds. Following this change we re-done all

our analyses and elaborate about the calculation of this variable in the Methods (lines 405-409).

Comment 22: L286: remove common. Eg *O. lugens* is not a common Palearctic bird...

Our response: Done.

Comment 23: L289: Remove “producing a 289 dataset of 101-491 individuals per species (mean 211 individuals; table S1).”

Our response: Done.

Comment 24: L290: straightforward

Our response: Corrected.

Comment 25: L291: generation

Our response: Corrected.

Comment 26: L310: I think you need to use generalized linear models because the moult score is a Poisson variable.

Our response: We note that following the change in the data (as explained in Comment #21), the distribution of the data was found to be a gamma distribution. Following this comment, we re-analyzed our models using GLM as suggested by the reviewer.

Comment 27: L312: I think model selection in this context is totally unnecessary. You have only a couple of explanatory variables (temp and sex), both with well-defined a priori hypothesis of their effect. Actually, you want to test both variables. Thus, I would carry out full models in all dichromatic species (i.e. temp*sex).

Our response: Model selection procedure is necessary in our case because moult extent may or may not be affected by GMTA, bird sex and their interaction and consequently we must select the appropriate model among the different options and the full model may not be the most appropriate in all cases. We specifically note that the interaction may affect the variance structure of the model, and consequently including or excluding it from the model may substantially affect the results. For example, running the full model on *Oenanthe hispanica* could have led us to using a model that is outperformed by the

currently selected model that includes only GMTA and sex but not their interaction (see table S3 and S4).

Comment 28: Fig1. Circle insets are misleading in their current position. They should be placed in the boxed inset. The moult pattern under warm temperatures shown in the right circle of panel A looks really weird. Postjuvenile moult always involves coverts and inner secondaries. Pictures of C are really nice, but the difference between juvenile and adult feathers is excessively exaggerated. Unfortunately, differences between juvenile and adult feathers are subtler than that depicted.

Our response: We suggest that this figure is very informative. Specifically, the lines between circle insets and the trend-line represent the effects of different climates on the trend-line and these climates are connected using a line to the circle insets. The moult pattern under warm temperatures shown in the right circle inset of panel A is actually a regular moult pattern for many species (for example, Jenni & Winkler, 1994; Gargallo and Clarabuch, 1995, Extensive moult and ageing in six species of passerines (Ringling & Migration 16 (3) 178-189) and Shirihai et al., 2001, *Sylvia* warblers: identification, taxonomy and phylogeny of the genus *Sylvia*). Post-juvenile moult may be extensive and may include more feathers than the wing-coverts and inner secondaries, especially in southern species or when the moult takes place after the autumn migration (*e.g.*, in *Lanius isabellinus*, *Lanius senator*, *Lanius excubitor* and rarely also in *Turdus merula* and *Ficedula albicollis*; for example see Figure 1 below).

Figure 1C – Thank you. We suggest that the difference between juvenile and adult feathers in this figure reliably reflects the reality. We note that in the present study we examined only species that are characterized by a clear, contrasting, and easily recognizable difference between moulted and non-moulted feathers. We provide in the supplementary material a figure that includes a photograph of a partially moulted wing, which demonstrates the differences between moulted and non-moulted feathers in one of our studied species (*Ficedula albicollis*; Figure S3). Either way, Figure 1C was devised to highlight differences between moulted and non-moulted feathers and not to display the exact differences between these two types of feathers. By adding the photo of the wing to the supplementary material we now provide the readers with the opportunity to judge these differences by themselves.

Figure 1. Southern Grey Shrike *Lanius (excubitor) aucheri*, after extensive post-juvenile moult. The box bottom-right displays the moulted (dark-grey) and unmoulted (white) feathers – the moult includes wing-coverts, tertials, inner secondary and outer primary feathers.

Comment 29: Table S1:

- Is *Lanius isabellinus* referring to the former complex of subspp (currently split into: *L. isabellinus*, *L. phoenicuroides*, *L. speculigerus*) or to the new species (former subspp *L. i. isabellinus*, *L.i. tsaidamensis*, and *L.i. arenarius*).
- Migration distance seem a bit excessive for some spp, such as *O. deserti*, *M. solitarius* or *T. merula*. Note, for instance that *M. solitarius* is migrant only in the Caucasus region. *T. merula* is rather a partial migrant, i.e. only part of individuals moves.
- Sexual dichromatism is hard to distinguish in juveniles of *O. hispanica*, *F. hypoleuca* and *L. senator*.

Our response: *Lanius isabellinus*: *Lanius isabellinus* refers to only two subspecies that migrate through the Western-Palearctic (*isabellinus* and *phoenicuroides*). Following this comment we added a row in the updated Table S1 which lists the subspecies tested in each species.

Migration distance: following this comment we examined again the migration distance, also using the maps provided in the Handbook of the Birds of the World Alive (del Hoyo et al., retrieved January 2019). The distance for *Monticola solitarius* and *Turdus merula*

was slightly shortened, but this was not the case for *Oenanthe deserti* in which the distance remained the same. The migration distance was calculated as the distance between the mid-breeding and the mid-wintering areas of each species. Thus, for species that perform only a partial migration the non-breeding population may influence the distance but in general the species is still considered migratory.

Sexual dichromatism: Sexing of juveniles *Oenanthe hispanica* in autumn, right **after their post-juvenile moult**, is not a difficult task (see, for example, Svensson et al., 1999, Collins Bird Guide, pp 285). In spring, it is even more straightforward. In the present study we included *Ficedula hypoleuca* and *Lanius senator* juveniles, only after their winter moult, when their sex can be determined with certainty, similar to adult birds.

Comment 30: Table S3: In a simple t-test, the df would be 366. Maybe you need to provide more details about this analysis. Moreover, see my comment above about non-normality of moult score.

Our response: Following this comment we re-examined our statistical test and changed the test to the non-parametric Mann-Whitney U-test. We note that the results (a significant difference between southern and northern sub-species) did not change following this change in the applied test.

Comment 31: In the reporting summary, you state that the study did not involve field work. This disagrees with the statement of L270 as well as with your statement about author contribution.

Our response: We used a large moult database that was collected during field work, mainly by Yosef Kiat, over many years and not specifically for the present study. Following this comment the relevant text in the Methods and the author contribution statement was revised.

Comments by Reviewer #2:

This is an interesting and timely study that utilizes natural history collections (NHCs) to look at changes in molting in juvenile birds over the past 211 years and whether this can be linked to climate change. This paper helps to advance our knowledge about how species are responding to climate change.

However, I do have several concerns. Neither the methodology nor the analysis included enough details. Consequently, I have a hard time understanding the underlying data structure and evaluating whether the appropriate statistical technique was used. I also have some concerns about the inferences made by the authors based on the results they found.

Our response: We hope that we provide sufficient information about our methodology and clarify the results in the updated version of the manuscript to support our inferences. Please find below our response to your specific comments.

Methodology

Comment 32: What were the localities associated with these specimens? A map would be very helpful. It was difficult to get a sense of the extent of geographic variation present in the data.

Our response: See our response to comments #1 and #2. The specimens were collected and measured from many different places in the Western Palearctic region. Yet, the birds have, likely in most cases, moved from their moulting localities to the collection and measurement localities and consequently their exact moulting locations are unknown. We believe that the many different sources of each collection, the inclusion of 10 natural history collections and the field measurements database from four different countries, and the movement of the birds between their moulting and collection/measurement sites, approximate a random sampling of the study species within the Western-Palearctic region in our database.

Comment 33: What are the dates of collections of these specimens? How much did they vary within a year? This could be problematic based on the potential mechanisms related to phenology and timing.

Our response: The date of bird collection during the year is not expected to affect our results because the condition of the feathers does not change between two moults that could have an interval of up to 11 months in some cases. Please see our detailed response to comment #2.

Comment 34: Is the locality associated with the specimen the same as the geographic moult location (line 299)?

Our response: No, most birds likely moved during the period between the end of their moult until they were collected/measured in the field. This information is provided in the second part of this sentence (lines 433-435 in the updated manuscript).

Comment 35: Why was field data added to the collections, what years and localities?

Our response: We combined data that were collected in the field with data from museum collection because field data are not available for the earlier years in our study period and the collection of museum specimens became rare in later years during the study period. However, the difference between these two data sources is not expected to influence our results. We chose species in which the moult is easy to identify, both in skins and in the field, and without reading the label attached to the specimen there is not chance to estimate the year of collection because well prepared and preserved skins that could be 200 years old look like a recently prepared skins or like live birds in the field. See also our response to Comment #28.

Influence of temperature

Comment 36: The authors didn't provide enough information about the role of temperature in this molting process, particularly at the scale of this study. At what spatial and temporal scale do temperature conditions influence molting? Do temperatures at the molting areas affect the extent of molting? The authors discuss two potential mechanisms by which temperature can influence molting- via time or resources. Are there any direct effects of temperature (i.e. physiological) on the process of molting, for example mediated through changes in developmental rate?

Our response: We suggest that the changes we documented in the extent of moult are indirectly caused by either a longer duration of the period during which moult takes places within the birds' annual cycle, or by the more abundant food resources available for moult. We did not find any empirical evidence in the scientific literature for direct physiological effects of temperature variation on moult extent. However, in the future, we do plan to test if such direct physiological effect on moult extent exists.

Comment 37: Why were Global Mean Temperature anomalies used? How were the anomalies calculated? What about geographical variation in temperature change? Are the authors arguing that geographical variation in temperature change is not important i.e. a

cold year is a cold year no matter where you are? There needs to be more clarity about the ecological relevance of this variable and the potential limitations of using it.

Our response: Regarding the reasons for using GMTA instead of a local index see our response to comment #5. Indeed, there is high correlation between several indexes we considered and therefore using different indexes is not expected to considerably affect our results (mean Pearson correlation coefficient is 0.91; see Table 2 above that follows comment #5).

Temperature anomalies indexes are regularly used in climate studies as a basic method for indication how warmer or colder the climate is in a particular place and time compared to the mean. According to the data provider (NASA), in the case of GMTA, the anomaly values were calculated relative to the mean temperature of the 1951-1980 reference period. This information was added to the Methods (lines 443-444). Yet, we note that the trend does not depend on the choice of the reference period. For example, if the mean temperature in a particular year is one degree higher than a year ago, so is the corresponding temperature anomaly, no matter what the reference period is, since the mean temperature used as a reference is the same for both years. We note that GMTA, by its nature, is an index that cannot capture small-scale variation over local spatial scales and within short-time scales (e.g., between different months). Therefore, the ecological relevance of this index is limited when inferring different biological processes that take place over smaller spatial and temporal scales in which fine-scale variation is important (lines 445-449).

Comment 38: There is almost certainly greater uncertainty surrounding the temperature measurements in the first time period. Are the results sensitive to the removal of this dataset?

Our response: The measurements that were used for estimating temperature variation in the first time period (<1880) are reliable and were used in many studies, including those by Hawkins et al. (Bulletin of the American Meteorological Society, 2017), Juckes et al. (Climate of the Past, 2007), Huang et al. (Nature, 2000), Frank et al. (Nature, 2010), Mann et al. (PNAS, 2008), Esper et al. (Science, 2002), Xing et al. (Plos One, 2016) and many more). Either way, the results are not sensitive to the removal of the 1805-1879 data period (see Table 3 below).

Table 3. Analysis results after removal of the 1805-1879 data period.

		r^2_{model}	Coefficient	GMTA	
				t	P
Oenanthe oenanthe	~ GMTA	0.03	78.18 ± 37.88	2.06	0.04
Oenanthe deserti	~ GMTA	0.09	160.57 ± 43.11	3.72	< 0.001
Oenanthe hispanica	~ GMTA	0.03	45.41 ± 19.38	2.34	0.02
Oenanthe pleschanka	~ GMTA	0.01	43.75 ± 28.56	1.53	0.13
Oenanthe lugens	~ GMTA	0.06	90.01 ± 38.73	2.32	0.02
Oenanthe leucopyga	~ GMTA	0.08	223.37 ± 91.92	2.43	0.02
Monticola solitarius	~ GMTA	0.08	225.88 ± 89.43	2.53	0.01
Turdus philomelos	~ GMTA	0.04	61.57 ± 14.45	4.26	< 0.001
Turdus iliacus	~ GMTA	0.05	69.12 ± 17.80	3.88	< 0.001
Turdus viscivorus	~ GMTA	0.00	-15.97 ± 48.09	-0.33	0.74
Turdus pilaris	~ GMTA	0.06	118.54 ± 29.18	4.06	< 0.001
Turdus merula	~ GMTA	0.02	1493.65 ± 229.16	6.52	< 0.001
Turdus torquatus	~ GMTA	0.01	38.68 ± 29.67	1.30	0.19
Ficedula hypoleuca	~ GMTA	0.00	-5.44 ± 35.03	-0.16	0.88
Ficedula albicollis	~ GMTA	0.07	161.73 ± 70.32	2.30	0.02
Lanius isabellinus	~ GMTA	0.02	316.30 ± 264.30	1.20	0.24
Lanius senator	~ GMTA	0.17	627.90 ± 106.84	5.88	< 0.001
Lanius e. excubitor	~ GMTA	0.03	131.13 ± 72.65	1.81	0.07
Lanius e. aucheri	~ GMTA	0.12	1050.70 ± 276.50	3.80	< 0.001

Analysis

Comment 39: Were the assumptions of linear (parametric) models evaluated and met?

Our response: No parametric analysis was used in the updated version of the manuscript.

Please see our responses to comments #26 and #30.

Comment 40: By modelling each species individually, the authors are assuming each species is responding differently to temperature. This could overstate the degree of interspecific variation and influence the estimate of central tendency. Did the authors consider using mixed-effects models? Also concerning is that the uncertainty surrounding the relationship between GMTA and molting extent for each species will vary given the differences in sample size and number of years sampled across species. Was this taken into account?

Our response: Indeed, we assumed that each species may potentially respond differently to temperature. Following this comment we tested a GLMM for all the species included in our analysis (species was included as a random factor) while also considering the differences between sample sizes using weights. We found that the results of the GLMM regarding the effect of GMTA on the moulted feather area is significant (Coefficient = 0.13 ± 0.002 , $t = 57.47$, $P < 0.001$). We note though that using GLMM is incorrect because closely related species cannot be considered statistically independent from each other (see also our detailed response to comment #43 below), and hence using a random factor for the different species does not address the problem of phylogenetic non-independence. Regarding the temporal distribution of the data, see our response to comment #3 and Table 1 above that follows Comment #3).

Comment 41: How was the general trend and its CI evaluated? What is the effect size?

Our response: The information regarding the general trend is given in our response to comment #40. Following this comment we added the coefficient of determination (r^2), a measure of effect size, to the supplementary material (Tables S3 and S4; calculated using the R package 'rsq' (Zhang, 2018)).

Comment 42: Why was the effect of migration distance and moult strategy tested?

Our response: These two variables were tested because they are known as factors that may affect the time available for moulting and hence they may affect the extent of feather moult (Ginn & Melville, 1983; Jenni & Winkler, 1994; Kiat et al., 2018). Hence, we tested if the response of bird feather moult to temperature variation may be additionally influenced by these factors. We note, however, that no such effect was found (see Table S5). Following this comment we revised the relevant text (lines 460-462).

Comment 43: Did a PGLS model better fit the data than a GLS? To me it's not clear whether a PGLS model was needed. Why was Brownian motion the only hypothesis considered?

Our response: Yes, PGLS model is a better fit than GLS. Since species-specific traits are known to be phylogenetically conserved, closely related species cannot be considered statistically independent from each other. Thus, we used PGLS to account for phylogenetic non-independence, instead of using a GLS which assumes that each data-point is independent (we explain this point in the Methods, lines 464-468). For a comprehensive review of relevant comparative methods and their use in ecological studies see Mason (2010), Felsenstein (1985) and Freckleton et al. (2002).

Mason, G. (2010) Species differences in responses to captivity: stress, welfare and the comparative method. *Trends in Ecology & Evolution*. 25(12): 713-725.

Felsenstein, J. (1985) Phylogenies and the comparative method. *The American Naturalist*. 125(1): 1-15.

Freckleton et al. (2002) Phylogenetic analysis and comparative data: a test and review of evidence. *The American Naturalist*. 160(6): 712-726.

Results

Comment 44: Rather than reporting a range of t-values across species (e.g. line 44), authors could tally up the number of species with a significant fit.

Our response: The number of species with a significant fit is mentioned earlier in the same sentence. Following this comment we deleted the t-values from the main text.

Comment 45: Where are the supporting results (i.e. species-level) for the temporal trend analysis?

Our response: We added another table in the supplementary material with results of a temporal trend analyses for the different species (see Table S4).

Data presentation.

Figure 1.

Comment 46: Authors need to explain how to interpret the temperature axis.

Our response: Following this comment we added this information to the legend of the figure. Please, also see our response to comment #37.

Comment 47: In the inset figures, the authors have not shown the underlying data supporting the general trend.

Our response: The information was added.

Comment 48: Caption for 1B is confusing because the authors show relationship with temperature AND across time but only mention the temporal trend in the caption.

Our response: We revised the legend of this figure and added the information regarding the relationship with temperature.

Figure 2.

Comment 49: I find this Figure confusing. Do parts B-D directly relate to the references cited in the caption? Or do the authors infer temporal changes in the annual events based on the results from the previous studies? This balance needs to be better articulated. Are there estimates of the time period of moulting available for each year? If so, why wasn't this directly related to changes in molting extent?

Our response: The changes in the timing of annual routine events which are illustrated in the lower part of the figure (panel B in the updated version of the manuscript) are largely based on published studies. Estimates of the time period of moulting for each year are unfortunately unavailable. Following this comment we revised the legend of the figure.

Comment 50: How was the moult index calculated? Proportion of which trend?

Our response: Following the use of a different measure of moult extent (feather cover area rather than the moult index that was previously used), the results of the analysis and several comments by the reviewers, we revised this figure and its legend. In the updated version of the figure, the moult index represents the proportion of the mean moult extent, among all species from which data is available at that year, for each year. The values of the moult index thus represent values between the minimum mean moult extent (index value = 0) and the maximum mean moult extent (index value = 1).

Conclusions

Comment 51: The authors conclusions about the differences between sexes are overstated (e.g. 15-18, lines 49-50, line 52-54, line 128-129). The authors only found a difference between sexes for 5/10 species. This is only half the time. Also, finding a

relationship between molting extent and GMTA does not necessarily mean that there's a trend through time, as the authors infer in lines 52-54.

Our response: We revised and improved the text in all the four sections that were mentioned by the reviewer. The findings suggesting a relationship between moult extent and climate warming does mean that there is a trend through time, because climate warming is highly correlated with time, and we clarify this point throughout the updated manuscript (see also comment #12 by reviewer #1 and our response).

Comment 52: The conclusions about the role of temperature is also overstated in some places (e.g. line 41 and title of paper). I don't think the results following the analyses conducted in this paper demonstrate that climate change has changed bird appearance, as the title suggests. The authors showed directional change in proportion of feathers molted over time and showed that there were correlations with inter-annual variation in temperature (again assuming I understand what GMTA is capturing based on the details provided). Specifically, the authors showed that warmer years lead to more extensive molting (assuming that positive values of GMTA = warmer years, see comment about Figure 1) but I'm not convinced they showed that directional changes in temperature over time led to these changes in bird appearance.

Our response: We showed that warmer years lead to more extensive moulting, which is highly correlative also with year (Figs 1 and 2). As we note throughout manuscript, juveniles that replaced their nest-grown feathers gain the ornamental appearance of adult plumage following their post-juvenile feather moult (e.g., line 15-16 and lines 38-40), so that a more extensive moult makes their appearance more adult-like. As we mention in our response to Comment #4, the extent of moult is strongly and positively related to year, but GMTA better explains moult extent than year (see also the detailed statistical results in Tables S3 and S4). We also note that, unsurprisingly, year and GMTA are positively correlated (Huang et al., 2000; Esper et al., 2002; Juckes et al., 2007; Mann et al., 2008; and Frank et al., 2010).

Minor comments

Comment 53: line 8: The first I read this through I wasn't sure what the authors meant by 'individual appearance'. Because this journal has a general audience, it would help to clarify what you mean here. For example it could also be interpreted to mean their first appearance within a season.

Our response: We revised this sentence and clarified this point (lines 15-16).

Comment 54: What plumage characteristics were used to determine the age of the bird? Is there any uncertainty with identifying first year juveniles? Are adults any bird older than first-year?

Our response: We identify the age based on the unique characteristics for each species that were published by Svensson (1992, Identification Guide to European Passerines) and Jenni & Winkler (1994). Regularly, juvenile nest-grown feathers are poorer in texture, duller and characterize by higher abrasion level than that of feathers growing in the post-juvenile moult or by adults. Juvenile birds following their partial post-juvenile moult are characterize by a mixture of nest-grown and post-juvenile feathers. In contrast, adults (always older than one year) of the study species have only a single generation of feathers. Therefore, there isn't any uncertainty with identifying first-year individuals. We added this information to the manuscript (lines 391-395).

Comment 55: I think it would be helpful to have a diagram of the anatomy of the feathers showing which parts were assessed for molting extent.

Our response: Done (see Figure S2).

Comment 56: several typos (e.g., lines 29, 34)

Our response: Corrected.

Comment 57: In reading the introduction, my question immediately was- what are the implications if juveniles look more like an adult? This is addressed in lines 114-125. I would suggest moving this section to the introduction as it helps to better set up the study.

Our response: Done. Following this comment we added this information to the Introduction (lines 43-44).

Comment 58: line 34- what differences about the sexes would lead to differences in thermoregulation and behaviour (e.g., size? plumage?)

Our response: Both examples are relevant since size-related and chromatic (plumage) characteristics may predict how each sex may be differently affected by climatic conditions. This information is presented in lines 152-154.

Comment 59: Table S2- The coefficient of the interaction term for those species where there was a significant interaction should be included.

Our response: Done (Table S3 in the updated version of the manuscript).

Comments by Reviewer #3:

This is a well-executed manuscript showing that the extent of the post-juvenile (preformative) molt has become more extensive over the past 200+ years among 19 European bird taxa. Text could stand a bit of copy-editing (e.g., Summary paragraph, 7th and 10th lines) but length is appropriate, methods and statistical procedures seem sound, sample sizes are excellent, figures are well-designed and informative, and supplementary materials are adequate. Findings are novel and will be of interest to students of bird molt and those tracking the effects of global climate change.

I have a few rather minor specific comments:

Comment 60: The term "juvenile" is widely used to refer to a bird in juvenile plumage. It then undergoes a preformative (or "post-juvenile") molt during the first 1-6 months of life, after which it is in formative (post-juvenile) plumage. A sentence summarizing this and these terms would be useful early in Main text to put the study in context (replace 3rd-4th sentences). Throughout ms. carefully construct wording accordingly. E.g., "...causes juveniles to look more like adults..." (Summary paragraph, line 11) is incorrect as it is the resulting formative plumage that will more-closely resemble that of adults (in some species) if the preformative molt is more extensive. "...juvenile moult" (e.g. Fig. 1 caption) should be replaced by post-juvenile or preformative moult, etc.

Our response: We added the information suggested by the reviewer to the first paragraph of the manuscript (lines 17-20) and also improved the moult terminology in Fig. 1 legend and throughout manuscript (for example, lines 21, 76, 148-149, 193 and 420-421).

Comment 61: Main text, 4th sentence: Point out that the preformative molt can vary widely in extent on both an individual and taxa-specific context (as related to available time and resources).

Our response: Done.

Main text

Comment 62: 1st paragraph, last line: Point out that these two previous studies examined timing of molt. This study is the first to examine extent to my knowledge.

Our response: Thank you. Done.

Comment 63: 2nd paragraph: Establish here that the study species breed in (northern?) Europe and show various migration strategies. [Since effects of global climate change on bird molt will vary widely around the globe, it is important to put this study into a boreal-centric context at the outset.]

Our response: Done (line 53-54).

Comment 64: 3rd paragraph, 1st line: Recent evidence indicates that most post-breeding and post-juvenile molting does not occur "at the breeding grounds" - better to say "near" the breeding grounds or within the breeding range of the species.

Our response: Corrected.

Comment 65: Three taxa show no migration and are thus "resident" (Table S1). It could be worthwhile to separate these. Related, it would seem earlier breeding (II) would occur in all species, not just the migrant species.

Our response: We suggest not to separate between residents and short-distance migratory species as both are wintering in the Northern Hemisphere, an attribute that largely determine their moult timing and the time available for each annual-cycle event. The information about the earlier breeding timing was corrected, thank you (lines 98-99).

Comment 66: 5th paragraph - Females of course also differ in their breeding biology, in many species resulting in later molt after more-extensive nesting and post-fledging responsibilities than males.

Our response: This information was added to the manuscript (lines 145-151). See also our response to comment #16.

Comment 67: The argument that the sex-specific difference is related to feather melanin does not hold on its own. Authors would first need to demonstrate that males had molted fewer feathers than females before climate-change effects were incurred. I know of no study that demonstrates this. Alternatively, there could be an effect if more melanin is

produced later during the post-juvenile molt in males. This effect (termed "molt-plumage interaction" - Pyle 2013a) is seen in North America, particularly among long-distance migrants such as orioles and *Pheucticus* grosbeaks, whereby feathers replaced later during the molt show more adult (definitive) characters as the timing of molt advances, including the production of melanin. Do all five species showing this sex difference have sex-specific plumage involving melanin and, conversely, do the five species not showing the sex difference lack extensive feather melanin? This would be good to know. In general I think the melanin connection can be mentioned but not emphasized, and that other explanations related to male-female differences in behavior, e.g., timing of molt related to breeding, should be proposed. [Pyle, P. 2013a. Dark-faced Common Murres off California in fall and winter. *Western Birds* 44:250-261]

Our response: We suggest the opposite, that before climate-change effects were incurred, males moulted more feathers than females, but today, when there is more time available for moulting, females moult more feathers than males. Yes, all species showing this sex-related difference in moult extent have sex-specific plumage involving melanin and, conversely, these species not showing sex-related difference in appearance lack extensive feather melanin. Following this comment we now include a new figure in the updated manuscript (Figure 3) that highlights this difference between these two groups of species following an analysis of male plumage ornamentation (see also lines 158-164). We added the explanations related to male-female differences in behavior in lines 145-151 (see also our response to comment #16).

Comment 68: Sixth paragraph: Molt extent has been proposed as being very plastic due to extensive variation both within species and between closely related species that, e.g., demonstrate differing migration strategies (Pyle 2013b). [Pyle, P. 2013b. Evolutionary implications of synapomorphic wing-molt sequences among falcons (Falconidae) and parrots (Psittaciformes). *Condor* 115:593-602]

Our response: Following this comment we added the suggested reference to the relevant sentence (line 191).

Comment 69: Final two lines. Whether or not the preformative molt can vary to complete is not a linear function in passerines and varies phylogenetically. I doubt that most of the study species (e.g., any of the Turdidae) will shift to having a complete preformative molt (i.e., including all remiges) as a response to climate change (Turdidae

are relatively fixed in molting variable numbers of upperwing coverts but few if any primaries). Recast this simply in terms of more feathers replaced with time. Further, complete preformative molts being "a common pattern in tropical species" is true to a minor but not a major extent, at least in Neotropical passerines (Johnson and Wolfe 2018), and it would appear to have more to do with solar exposure among species than anything to do with temperature. Thus, the implicit connection between global warming and molt extent with this statement should not be suggested. [Johnson, E. I., and J. D. Wolfe (2018). Molt in Neotropical birds: Life history and aging criteria. *Studies in Avian Biology* 51]

Our response: Thank you. Following this comment we revised these sentences. We deleted our suggestion of complete moult among species that are largely affected by warming climate (lines 191-196).

Methods -

Comment 70: Second paragraph - "...northern species molt more extensively than southern species..." I do not believe this to be true in any context (e.g., see above re tropical vs. boreal species) but even within temperate zones the opposite seems more the case, at least in North America. Remove this as it has little bearing on this ms.

Our response: Thank you for recognizing our mistake. What we meant to write was that southern species moult more extensively than northern ones, of course. This sentence is now corrected (lines 431-433).

Reviewers' comments:

Reviewer #1 (Remarks to the Author):

I congratulate the authors for an excellent review of its manuscript. They have solved most of the questions raised during the first review round by adding a comprehensive reply to my comments as well as by adding more information in the manuscript. In spite of the fact that the current version looks more satisfactory, I have still a number of comments. I am pretty sure that the authors are aware about most of them. For this reason, I would like to see more recognition of these issues in the ms.

I will start by replying some of your comments from your letter.

Comment#3 “[...] we note that excluding data from earlier years is a wrong practice and data from all available sources should be included in the analysis [...]”

REPLY: I did not want to suggest to exclude some of your data. I just tried to highlight that you should not compare coefficients from regression models (i.e. slopes), when the length of time-series is notably different (e.g. fieldfare vs pied wheatear). These coefficients are simply providing info for different periods and thus, differences between them are to some extent meaningless.

Comment#4 “[...] Due to collinearity of GMTA and year (the correlation between them is $r = 0.92$, $P < 0.001$), we could not include these two factors together in the same models. [...]”

REPLY: I agree. Frankly, I could not imagine a so high collinearity between both variables. This value implies an almost perfect monotonic increase of globe temperature during the past 211 years. However, this is not true at all. For instance, temp did not increase at all neither during the XIX century nor between 1940s-1970s (see IPCC reports). In addition, looking at your fig.2A, I think there is too much scatter for a $r=0.92$. Anyway, my previous concern about GMTA effect on moult still remains in light of your response. With such a perfect temporal trend of GMTA, any other variable (such as moult) with a temporal trend (table S4) will show a correlation. You can detrend both time-series before to include them in the GLMM. As I stated before, I think that the observed trend on moult patterns is a really remarkable result, whatever the cause is. The authors did a good job in their models. The issue is on data characteristics.

Comment#27 “[...] Model selection procedure is necessary in our case because moult extent may or may not be affected by GMTA, bird sex and their interaction and consequently we must select the

appropriate model among the different options and the full model may not be the most appropriate in all cases. [...]"

REPLY: I still disagree with your approach. Model selection is especially suitable when you have a large number of predictors. Then, it can be useful to select only those model/s with the best balance between fitting and amount of variables (ie the most parsimonious solution). Here, you have only 2 predictors and 4 possible models. You have a priori hypotheses and predictions for all variables and you want actually to test all effects (ie, you want to know the effect of GMTA, the effect of sex, and how both effects interact). The only way to do this, in my opinion, is by a full model. As you can see in table S3, coefficients do not change dramatically between the full model and the simplified models (the only exception is in the p- values of *Oenanthe hispanica*). In your context, the magnitude of the parameters is not very essential. To me, the most important is their sign (to understand how affect each variable) and their significance (to know which effects are relevant and which not at all).

Nevertheless, if you persist in using model selection, you should use a fully coherent approach. In table S3, you cannot show results from an information theoretical approach (eg AIC values) and results from a hypothesis testing approach (eg p-values). In an information theoretical approach, p-values does not make sense. You should use Akaike weights to determine the relevance of the predictors. In addition, you should use multimodel averaging with those models with a $\Delta AIC < 2$. For instance, in *Lanius senator*, GMTA * sex and GMTA + sex can be considered highly probable, and even GMTA model can be considered probable as well.

I have still three general remarks to the new version of your manuscript:

1) Sexual differences: The authors put still too much emphasis on the sexual differences. Indeed, they specified that such differences were found only in 5 species. Thanks. However, they still spend lot of text to develop this idea (L66-72; L157-163). For instance, in L66-72, there is 6 lines devoted to differences, while just one short sentence to the lack of them. I realize that differences are more interesting than lack of them, but this fact cannot justify a so disproportionate attention. Actually, you have 10 dichromatic species and you found that females responded more than males only in half of them. Why is not the other half of your sample so important? Furthermore, looking at table S3, I think the magnitude of the effects of GMTA:sex interaction are modest compared to GMTA and sex (eg most of $p > 0.02$). In sum, I agree these sexual differences are an interesting result, but you should realize that it is only valid for half of a small sample of species ($n=10$) and consequently, you should focus less attention to this finding.

2) Amount of melanins: You propose that in dichromatic species with a GMTA:sex interaction significant, such interaction is different due to a differential degree of plumage melanisation between males and females. I agree this can be a suitable hypothesis. However, your evidence to support it is very weak. Your hypothesis is based mainly on regression shown in Fig. 3. With $n=10$ and $p=0.047$, I would be more cautious suggesting theories. Furthermore, see my comment below about Dale et al.'s scores. On the other hand, melanins represent <2% of feather mass (e.g. see Haase et al. *J. Heredity* 1992; 83:64 // McGraw et al. *Funct. Ecol.* 2005; 19:816), so I can hardly imagine how this fact can be a real constraint to produce a feather. In fact, melanins can be costly for the individual, but this fact can affect simply the colour of the feather rather than its full synthesis (which is 99% keratin). Finally, if your theory is valid, you can test it with all your species. I would expect that GMTA effect (or year trend) should be smaller in those more blackish species because they have more melanins in their feathers.

3) Random sample: You claim that your sample of individuals are a random sample of individuals coming from the full distribution range (eg L447). Actually, you have used several museum collections and life birds from several countries. However, you can simply assume that your birds are a random sample. As you recognize in your response letter, there is no information about the origin of most individuals. You can just trust that they are indeed a random sample, but you have no information to check the degree of geographical bias of your samples. Moreover, even if all specimens for a species are perfectly spread across all its distribution area, there is no guarantee of no biases. For example, if each museum collection was created during a particular time period and each collection focused mostly in some regions of the distribution area, then you will not have a random distribution of your samples along the time.

Some minor comments (L=line):

L22: I suggest: ...that is positively correlated...

L54: You studied actually 18 species. I suggest: ... moult in 19 passerine bird species...

L58: Fig. S1 is useful, but I think you do not need to cite it here.

L60: If you accept my suggestion for L54, you can remove passerine here.

L96: Would be more realistic to say "...during the winter"? During the autumn there is also lot of food due to fruiting of lot of plants. This is especially valid for Mediterranean latitudes, as the authors must know.

L102: I disagreed already with this sentence. Reduction in migration distance is not a phenological change. A reduction in migration distance can be the cause for a change in phenology. Note that earlier arrivals in spring can be also enhanced by closer wintering areas.

L111: I think there is no information about the arrival of LDM to their African quarters (but see Kok et al. Ardea 1991).

L140-4: This is a nice argumentation, but what about the species that make a complete post-juvenile moult? (eg larks, sparrows, starlings, long-tailed tit, etc). In addition, first year individuals can replace most of their juvenile plumage either just after fledgling (post-juvenile moult sensu stricto) or just before their first breeding season (pre-nuptial moult). In the first strategy, they spend the winter with an adult-like plumage, while in the second strategy they spend the winter with a juvenile plumage. I think that behavioural and physiological consequence may be pretty different for each moult strategy even if both can be seen as partial moults of the juvenile plumage.

L150-6: This is a nice theory added in response to my previous comment#16. However, according to your arguments, why do not females undertake a complete postjuvenile moult? If adult plumage is more durable during breeding season, then I would expect a directional selective pressure to have more extensive moults. This selection process would finish with a complete moult. This could not be the case of males, in which there is a trade-off between the benefits and costs of adult plumage. If a first year male has a more extensive moult, he can be more attractive to females, but he also can receive more aggressions by other males.

L161: I suggest: ... males moulted significantly more...

Fig. 3: You do not need to provide the PGLS in the legend again. Moreover, this result is the result of a simple linear regression model (I have redone the analysis with Dale et al's data and your data from table S3). P-value is 0.047. Dale et al. study is cool, but I am not sure whether their scores are

suitable for the present study with a few species. I have checked the differences in the scores between males and females in all your studies species. I have found some weird values. Dichromatic species had higher differences than monomorphic ones in general. However, *O. hispanica* (dichromatic) has smaller differences than *O. lugens* (monomorphic). *L. isabellinus* and *L. senator* are patently dichromatic, but they have a similar (and low) score to *L. excubitor* (a monomorphic species). Note that Dale et al. quantified those scores for adult plumages, while you are working with postjuveniles ones. Note also that Dale et al. quantified those score from plates of the HBW, which may imply some inaccuracies for particular species (eg complex patterns not well summarized).

L199: I miss here another sentence such as "If moult extent is showing a genetic change..."

References: Check scientific names, I think they should be in italics. Amend author names of reference 23. Check the "-" in references 24 and 51

L396-7: I think these pictures should be included in the table S2.

L458: I suggest: ...(n = 9; Table S1), for... In addition, maybe you should state n=8 (see my comment above).

L460: Please, specify the used link function.

Table S2: This table is useful and welcome. However, this information does not help to know the actual provenance of the samples. However, I agree with the authors that this is unknown. I miss in this table the individuals studied from pictures obtained in internet.

Table S3: It would be more informative to specify for which level of sex are the coefficients shown for "sex" and "GMTA:sex". I think they are for females. In addition, avoid <0.05; just give the exact p-value.

Table S7: I think the standard deviation would be a more suitable parameter of data dispersion.

Reviewer #3 (Remarks to the Author):

I have read through the thorough responses of authors to all three reviewer's suggestions and find the responses to my suggestions (and largely those of the other two reviewers) to be adequately addressed. I did not review the ms to see the changes but trust that they are as indicated.

Peter Pyle

ppyle@birdpop.org

Comments by Reviewer #1:

Comment 1: I congratulate the authors for an excellent review of its manuscript. They have solved most of the questions raised during the first review round by adding a comprehensive reply to my comments as well as by adding more information in the manuscript. In spite of the fact that the current version looks more satisfactory, I have still a number of comments. I am pretty sure that the authors are aware about most of them. For this reason, I would like to see more recognition of these issues in the ms.

Our response: Thank you for the constructive comments that helped us to improve our work. Please find below our responses to your comments.

I will start by replying some of your comments from your letter.

Comment 2: Comment#3 “[...] we note that excluding data from earlier years is a wrong practice and data from all available sources should be included in the analysis [...]”

REPLY: I did not want to suggest to exclude some of your data. I just tried to highlight that you should not compare coefficients from regression models (i.e. slopes), when the length of time-series is notably different (e.g. fieldfare vs pied wheatear). These coefficients are simply providing info for different periods and thus, differences between them are to some extent meaningless.

Our response: Thank you for clarifying this point. Following this comment we referred to this point in the Methods section by adding a note that the period over which the data was collected differed between the species and consequently one cannot compare between the GMTA coefficient values of different species (lines 281-284).

Comment 3: Comment#4 “[...] Due to collinearity of GMTA and year (the correlation between them is $r = 0.92$, $P < 0.001$), we could not include these two factors together in the same models. [...]”

REPLY: I agree. Frankly, I could not imagine a so high collinearity between both variables. This value implies an almost perfect monotonic increase of globe temperature during the past 211 years. However, this is not true at all. For instance, temp did not increase at all neither during the XIX century nor between 1940s-1970s (see IPCC reports). In addition, looking at your fig.2A, I think there is too much scatter for a $r=0.92$. Anyway, my previous concern about GMTA effect on moult still remains in light of your response. With such a perfect temporal trend of GMTA, any other variable (such as moult) with a temporal trend (table S4) will show a correlation. You can detrend both

time-series before to include them in the GLMM. As I stated before, I think that the observed trend on moult patterns is a really remarkable result, whatever the cause is. The authors did a good job in their models. The issue is on data characteristics.

Our response: As we noted in the response letter of our previous re-submitted manuscript version, AIC values suggest that GMTA is a better predictor of feather moult extent than year ($\Delta AIC = 35.0$). This strongly supports the inclusion of GMTA rather than the year as an explanatory variable in the analysis. In addition, $r = 0.92$ means that ~15% of the variance in the GMTA is not explained by year and this could result from periods over which GMTA did not increase, or increased only to a small extent, with time (e.g., 1940-70 that is mentioned by the reviewer). GMTA data were downloaded from the following website (<https://data.giss.nasa.gov/gistemp>) and we welcome independent inspection of the relationship between year and GMTA. Detrending a time series is usually done to facilitate the exploration of relatively weak or cyclic effects in the data by removing the strong effect of time. This is not the case here: (1) in models that include the year as a predictor, detrending the year variable is expected to result in no effect what so ever. (2) In models that include GMTA as a predictor, detrending the effect of GMTA using the year variable may result in the production of spurious results due to the high collinearity of year and GMTA because after taking the yearly trend away, the analysis will be made on a very small part of the variance of the data. Consequently, detrending the two datasets is a questionable procedure that should be avoided.

Comment 4: Comment#27 “[...] Model selection procedure is necessary in our case because moult extent may or may not be affected by GMTA, bird sex and their interaction and consequently we must select the appropriate model among the different options and the full model may not be the most appropriate in all cases. [...]”

REPLY: I still disagree with your approach. Model selection is especially suitable when you have a large number of predictors. Then, it can be useful to select only those model/s with the best balance between fitting and amount of variables (ie the most parsimonious solution). Here, you have only 2 predictors and 4 possible models. You have a priori hypotheses and predictions for all variables and you want actually to test all effects (ie, you want to know the effect of GMTA, the effect of sex, and how both effects interact). The only way to do this, in my opinion, is by a full model. As you can see in table S3, coefficients do not change dramatically between the full model and the simplified models (the only exception is in the p- values of *Oenanthe hispanica*). In your context, the

magnitude of the parameters is not very essential. To me, the most important is their sign (to understand how affect each variable) and their significance (to know which effects are relevant and which not at all).

Nevertheless, if you persist in using model selection, you should use a fully coherent approach. In table S3, you cannot show results from an information theoretical approach (eg AIC values) and results from a hypothesis testing approach (eg p-values). In an information theoretical approach, p-values does not make sense. You should use Akaike weights to determine the relevance of the predictors. In addition, you should use multi model averaging with those models with a $\Delta AIC < 2$. For instance, in *Lanius senator*, $GMTA * sex$ and $GMTA + sex$ can be considered highly probable, and even $GMTA$ model can be considered probable as well.

Our response: Thank you for the detailed explanation. We preferred using the information theoretical approach, instead of hypothesis testing due to the possible effect of the inclusion or exclusion of the interaction on the variance structure of the model, as explained in the response letter of our previous re-submitted manuscript version. Consequently, we completely accepted the reviewer's suggestions: (1) We excluded the hypothesis testing results from tables S3 and S4 by taking out the t and P values. (2) We calculated the Akaike weights for each tested model. And (3), we averaged the coefficients in cases of models with a $\Delta AIC < 2$ (see tables S3 and S4). Furthermore, since all our models include $GMTA$ (table S3) or year (table S4), we added another model that includes only a constant as a null model and added it to the list of models from which we selected the most likely model that explains feather moult extent (similarly, we also added a null model in the PGLS analysis, see table S5).

I have still three general remarks to the new version of your manuscript:

Comment 5: 1) Sexual differences: The authors put still too much emphasis on the sexual differences. Indeed, they specified that such differences were found only in 5 species. Thanks. However, they still spend lot of text to develop this idea (L66-72; L157-163). For instance, in L66-72, there is 6 lines devoted to differences, while just one short sentence to the lack of them. I realize that differences are more interesting that lack of them, but this fact cannot justify a so disproportionate attention. Actually, you have 10 dichromatic species and you found that females responded more than males only in half

of them. Why is not the other half of your sample so important? Furthermore, looking at table S3, I think the magnitude of the effects of GMTA:sex interaction are modest compared to GMTA and sex (eg most of $p > 0.02$). In sum, I agree these sexual differences are an interesting result, but you should realize that it is only valid for half of a small sample of species ($n=10$) and consequently, you should focus less attention to this finding.

Our response: As the reviewer mentioned, the sexual differences found in our work are an interesting result. We note that the magnitude of these sexual differences indeed varies between the species and the analysis that is illustrated in figure 3 indicates that these sexual differences are correlated with the level of sexual dichromatism. Therefore, our results explain both the significance sexual differences found in some species and lack thereof in other species. Yet, we agreed that our sample size is rather small and more study is required to further substantiate these findings. Following this comment we added a note regarding this point in the updated version of the manuscript (lines 172-175).

Comment 6: 2) Amount of melanins: You propose that in dichromatic species with a GMTA:sex interaction significant, such interaction is different due to a differential degree of plumage melanisation between males and females. I agree this can be a suitable hypothesis. However, your evidence to support it is very weak. Your hypothesis is based mainly on regression shown in Fig. 3. With $n=10$ and $p=0.047$, I would be more cautious suggesting theories. Furthermore, see my comment below about Dale et al.'s scores. On the other hand, melanins represent $<2\%$ of feather mass (e.g. see Haase et al J. Heredity 1992; 83:64 // McGraw et al. Funct. Ecol. 2005; 19:816), so I can hardly imagine how this fact can be a real constraint to produce a feather. In fact, melanins can be costly for the individual, but this fact can affect simply the colour of the feather rather than its full synthesis (which is 99% keratin). Finally, if your theory is valid, you can test it with all your species. I would expect that GMTA effect (or year trend) should be smaller in those more blackish species because they have more melanins in their feathers.

Our response: We found that in ten sexually dichromatic species, differences between the sexes in plumage ornamentation predict the variance in their sex-dependent response to GMTA, explaining $\sim 40\%$ of the variance. We thank the reviewer for pointing out important problems in the melanin cost explanation. Consequently, we made a thorough revision of this paragraph and deleted this explanation from the manuscript. It is not clear why the response to GMTA is influenced by sexual dichromatism, and we speculate that

two factors may be involved: (1) feather durability and (2) ornamentation plumage cost. Feather durability: feathers characterized by blackish and melanin-rich colour are more durable than non-melanic feathers (see for example, Averill 1923 and Bonser 1995), and hence, to overcome their higher feather wear, females may benefit from a more extensive moult compared with males, but only in species in which females have more lighter and therefore less durable plumage than males. Ornamentation plumage cost: this explanation is based on the costs and benefits of ornamented feathers of males. While these ornaments could provide mating advantages, they may also induce a cost, for example, if sex-biased mortality or a more aggressive behavior that is induced on these males occurs as a result of a more ornamented plumage (for example, Promislow et al. 1992, Senar et al. 1998 and Safran et al. 2008). Please see lines 162-172 for a discussion of these possible explanations. Please see also our responses to Comments 5 and 16.

[Averill, C. K. Black wing tips. *Condor* 25, 57–59 (1923); Bonser, R. H. C. Melanin and the abrasion resistance of feathers. *Condor* 97, 590 (1995); Senar, J. C., Copete, J. L. & Martin, A. J. Behavioural and morphological correlates of variation in the extent of postjuvenile moult in the Siskin *Carduelis spinus*. *Ibis*. 140, 661–669 (1998); Promislow, D. E. L., Montgomerie, R. & Martin, T. E. Mortality costs of sexual dimorphism in birds. *Proc. R. Soc. Lond. B* 250, 143–150 (1992); Safran, R. J., Adelman, J. S., McGraw, K. J., & Hau, M. Sexual signal exaggeration affects physiological state in male barn swallows *Curr. Biol.* 18, 461-462 (2008)]

Comment 7: 3) Random sample: You claim that your sample of individuals are a random sample of individuals coming from the full distribution range (eg L447). Actually, you have used several museum collections and life birds from several countries. However, you can simply assume that your birds are a random sample. As you recognize in your response letter, there is no information about the origin of most individuals. You can just trust that they are indeed a random sample, but you have no information to check the degree of geographical bias of your samples. Moreover, even if all specimens for a species are perfectly spread across all its distribution area, there is no guarantee of no biases. For example, if each museum collection was created during a particular time period and each collection focused mostly in some regions of the distribution area, then you will not have a random distribution of your samples along the time.

Our response: Thank you. Following this comment we deleted the words "created a largely randomly sampled dataset" from the Methods section (lines 254-260).

Some minor comments (L=line):

Comment 8: L22: I suggest: ...that is positively correlated...

Our response: Corrected.

Comment 9: L54: You studied actually 18 species. I suggest: ... moult in 19 passerine bird species...

Our response: Corrected. We note that there is no consensus regarding the separation of *Lanius excubitor* into two species (for more information see Lefranc 1997 and Harris 2010 & Olsson et al. 2010 that are mentioned in the Methods section). The geographic distribution, migration patterns, plumage colour and, importantly, moult strategies, of this species complex can justify a split into two species (see table S6), Great Grey Shrike and Southern Grey Shrike. Because of that and since we explicitly deal with this point in the manuscript (lines 236-241), we preferred leaving this separation into two species, Great Grey and Southern Grey Shrikes.

Comment 10: L58: Fig. S1 is useful, but I think you do not need to cite it here.

Our response: Corrected.

Comment 11: L60: If you accept my suggestion for L54, you can remove passerine here.

Our response: Corrected.

Comment 12: L96: Would be more realistic to say "...during the winter"? During the autumn there is also lot of food due to fruiting of lot of plants. This is especially valid for Mediterranean latitudes, as the authors must know.

Our response: Corrected.

Comment 13: L102: I disagreed already with this sentence. Reduction in migration distance is not a phenological change. A reduction in migration distance can be the cause for a change in phenology. Note that earlier arrivals in spring can be also enhanced by closer wintering areas.

Our response: Thank you. We agree that reduction in migration distance is not a phenological change and earlier arrivals in spring can also be enhanced by closer wintering areas but we could not find evidence for this effect in the spring migration of any species (in contrast to the autumn migration). Following this comment we revised the text in the updated manuscript version (lines 102-104).

Comment 14: L111: I think there is no information about the arrival of LDM to their African quarters (but see Kok et al. Ardea 1991).

Our response: Despite the general lack of information regarding the arrival to the tropical wintering areas, this is a logical result of an earlier departure from the breeding areas. Following this comment, we revised this sentence (lines 111-114).

Comment 15: L140-4: This is a nice argumentation, but what about the species that make a complete post-juvenile moult? (eg larks, sparrows, starlings, long-tailed tit, etc). In addition, first year individuals can replace most of their juvenile plumage either just after fledgling (post-juvenile moult sensu stricto) or just before their first breeding season (pre-nuptial moult). In the first strategy, they spend the winter with an adult-like plumage, while in the second strategy they spend the winter with a juvenile plumage. I think that behavioural and physiological consequence may be pretty different for each moult strategy even if both can be seen as partial moults of the juvenile plumage.

Our response: Juveniles of most Western-Palaearctic passerines do not perform a complete post-juvenile moult. Kiat & Izhaki (2016) suggested that this is largely a result of a time stress and the lower capabilities of juveniles to acquire necessary food resources for this process. Additionally, the intensity of the time stress inflicted on the birds may be influenced by several factors, including the species migration distance, breeding latitude, body size and diet. Moulting just before the first breeding season is overall rare and limited in its extent. Additionally, 1st year individuals that moult in their wintering areas mostly undertake feather moulting immediately after their arrival to the wintering areas. As mentioned in the manuscript, we explicitly tested whether bird moult strategy (pre- or post-autumn migration moult timing) affect its response to GMTA and found no such effect (lines 54-58). Following this comment we emphasize that this statement is valid only for Western-Palaearctic passerines (lines 139-141).

Comment 16: L150-6: This is a nice theory added in response to my previous comment#16. However, according to your arguments, why do not females undertake a complete postjuvenile moult? If adult plumage is more durable during breeding season, then I would expect a directional selective pressure to have more extensive moults. This selection process would finish with a complete moult. This could not be the case of males, in which there is a trade-off between the benefits and costs of adult plumage. If a first year male has a more extensive moult, he can be more attractive to females, but he also can receive more aggressions by other males.

Our response: We agree. We note, however, that females may still be time stressed (*e.g.*, by their northern breeding latitude and long migration distance; for example see Kiat & Sapir 2017) to an extent that may limit their ability to complete their post-juvenile moult and hence it is unknown whether the process will eventually result in a complete post-juvenile moult in females. The stronger response to GMTA of females compared to that of males in several species and the lack of a stronger response of males compared to that of females in any of the studied species, which are reported in the manuscript, may indeed indicate a directional selective pressure. Yet, we cannot currently establish whether this finding is due to phenotypic plasticity or natural selection, or both, and we hope that future studies will clarify this point.

Comment 17: L161: I suggest: ... males moulted significantly more...

Our response: Corrected.

Comment 18: Fig. 3: You do not need to provide the PGLS in the legend again. Moreover, this result is the result of a simple linear regression model (I have redone the analysis with Dale et al's data and your data from table S3). P-value is 0.047. Dale et al. study is cool, but I am not sure whether their scores are suitable for the present study with a few species. I have checked the differences in the scores between males and females in all your studies species. I have found some weird values. Dichromatic species had higher differences than monomorphic ones in general. However, *O. hispanica* (dichromatic) has smaller differences than *O. lugens* (monomorphic). *L. isabellinus* and *L. senator* are patently dichromatic, but they have a similar (and low) score to *L. excubitor* (a monomorphic species). Note that Dale et al. quantified those scores for adult plumages, while you are working with postjuveniles ones. Note also that Dale et al. quantified those

score from plates of the HBW, which may imply some inaccuracies for particular species (eg complex patterns not well summarized).

Our response: Following this comment we removed the PGLS from the figure legend. Because the maximum likelihood of λ value is 0.00 (we added this information to the updated manuscript) the result of the PGLS is the same as a simple linear model. We suggest that the scores devised by Dale et al. (Nature, 2015) are meaningful and we refer to the specific points that were mentioned by the reviewer: (1) *O. lugens* used in the study of Dale et al. is the subspecies known as Abyssinian Wheatear (*O. l. lugubris*) which is largely dichromatic and not monomorphic but the subspecies we tested (*lugens, persica*; Table S1) does not show any sexual dichromatism. We agree that differences between subspecies, such as in *O. lugens*, are not well summarized in Dale et al.'s study. Since we could not determine the sex of *O. lugens* individuals that are included in our study, this species was not included in the list of sexual dichromatic species in our study. Consequently, the analysis of this species in our work is not relevant for understanding sex-related responses to GMTA. (2) *L. isabellinus* and *L. senator* are not patently sexually dichromatic because sexual dichromatic plumage occurs in these species only in limited body parts and actually it is often not easy to identify the sex of the individual in these species, especially for non-specialist observers. Hence, it is reasonable that these species have similar sex-related differences in their ornamentation scores to those of *L. excubitor*. Regarding the reviewer's concern regarding the relevance of the scores devised by Dale et al. (2015) in relation bird age: the body feathers that were used to calculate the ornamentation score are replaced as part of the post-juvenile moult and consequently these feathers are similar to the feathers of adult birds, in these specific body parts.

Comment 19: L199: I miss here another sentence such as "If moult extent is showing a genetic change..."

Our response: Corrected (lines 194-196).

Comment 20: References: Check scientific names, I think they should be in italics. Amend author names of reference 23. Check the "--" in references 24 and 51

Our response:

Comment 21: L396-7: I think these pictures should be included in the table S2.

Our response: Data from pictures that were obtained from the internet were considered in the previous submission as field data. Following this comment we separated these data from the field data and provide it in a separate column.

Comment 22: L458: I suggest: ...($n = 9$; Table S1), for... In addition, maybe you should state $n=8$ (see my comment above).

Our response: Corrected. About the number of species see our response to comment #9.

Comment 23: L460: Please, specify the used link function.

Our response: The information was added (line 273).

Comment 24: Table S2: This table is useful and welcome. However, this information does not help to know the actual provenance of the samples. However, I agree with the authors that this is unknown. I miss in this table the individuals studied from pictures obtained in internet.

Our response: See our response to the comment #21.

Comment 25: Table S3: It would be more informative to specify for which level of sex are the coefficients shown for “sex” and “GMTA:sex”. I think they are for females. In addition, avoid <0.05 ; just give the exact p-value.

Our response: We added the information regarding the level of bird sex in the table. Following the revision of the table following Comment #4, the P values were excluded.

Comment 26: Table S7: I think the standard deviation would be a more suitable parameter of data dispersion.

Our response: Corrected.

Comments by Reviewer #3:

I have read through the thorough responses of authors to all three reviewer's suggestions and find the responses to my suggestions (and largely those of the other two reviewers) to be adequately addressed. I did not review the MS to see the changes but trust that they are as indicated.

Our response: Thank you.

REVIEWERS' COMMENTS:

Reviewer #1 (Remarks to the Author):

I have reviewed the manuscript and the response letter. The authors have overlooked a few issues:

- L268: You have not deleted "...created a largely randomly sampled dataset", as you replied in your letter. It is ok if you disagree in this point with my previous arguments about random origin of your data. I suggest the next rewriting: "Each museum collection used in the study consists specimens of various sources that were collected in different localities. By using data from ten bird collections and from live birds that were examined in several countries (Supplementary Table S2), it can be assumed that we created a largely randomly sampled dataset."

- Some minor typos are still remaining in the reference list. Check references e.g. 23, 30.

My final comment is to thanks the authors for their work and efforts during these three arduous review rounds. At least, I have enjoyed our discussions and disagreements. Hope the authors have the same thought in spite of the fact that I can be a tediously unforgiving reviewer sometimes. All my comments and criticisms have been always written with my genuine wish to improve your paper and promote your reflection. I like to see that most of them will appear finally in the published version as much as all your refutations to my comments.

Oscar Gordo [14-4-2019]

Comments by Reviewer #1:

Comment 1: L268: You have not deleted “...created a largely randomly sampled dataset”, as you replied in your letter. It is ok if you disagree in this point with my previous arguments about random origin of your data. I suggest the next rewriting: “Each museum collection used in the study consists specimens of various sources that were collected in different localities. By using data from ten bird collections and from live birds that were examined in several countries (Supplementary Table S2), it can be assumed that we created a largely randomly sampled dataset.”

Our response: Corrected following the Reviewer's suggestion.

Comment 2: Some minor typos are still remaining in the reference list. Check references e.g. 23, 30.

Our response: Corrected.

Comment 3: My final comment is to thanks the authors for their work and efforts during these three arduous review rounds. At least, I have enjoyed our discussions and disagreements. Hope the authors have the same thought in spite of the fact that I can be a tediously unforgiving reviewer sometimes. All my comments and criticisms have been always written with my genuine wish to improve your paper and promote your reflection. I like to see that most of them will appear finally in the published version as much as all your refutations to my comments.

Our response: Thank you for the constructive comments that helped us to improve our work.